# Defective Convolutional Layers Learn Robust CNNs

## Abstract

Robustness of convolutional neural networks has recently been highlighted by the adversarial examples, i.e., inputs added with well-designed perturbations which are imperceptible to humans but can cause the network to give incorrect outputs. Recent research suggests that the noises in adversarial examples break the textural structure, which eventually leads to wrong predictions by convolutional neural networks. To help a convolutional neural network make predictions relying less on textural information, we propose defective convolutional layers which contain defective neurons whose activations are set to be a constant function. As the defective neurons contain no information and are far different from the standard neurons in its spatial neighborhood, the textural features cannot be accurately extracted and the model has to seek for other features for classification, such as the shape. We first show that predictions made by the defective CNN are less dependent on textural information, but more on shape information, and further find that adversarial examples generated by the defective CNN appear to have semantic shapes. Experimental results demonstrate the defective CNN has higher defense ability than the standard CNN against various types of attack. In particular, it achieves state-of-the-art performance against transfer-based attacks without applying any adversarial training.

## 1 Introduction

Deep learning (LeCun et al., 2015), especially deep Convolutional Neural Network (CNN) (LeCun et al., 1998), has led to state-of-the-art results spanning many machine learning fields (He et al., 2016; Ren et al., 2015). Despite the great success in numerous applications, recent studies show that deep CNNs are vulnerable to some well-designed input samples named as Adversarial Examples (Szegedy et al., 2013; Biggio et al., 2013). Take the task of image classification as an example, for almost every commonly used well-performed CNN, attackers are able to construct a small perturbation on an input image. The perturbation is almost imperceptible to humans but can make the model give a wrong prediction. The problem is serious as some designed adversarial examples can be transferred among different kinds of CNN architectures (Papernot et al., 2016b), which means a machine learning system can be easily attacked even if the attacker does not have access to the model parameters.

There is a rapidly growing body of work on how to obtain a robust convolutional neural network, mainly based on adversarial training (Szegedy et al., 2013; Madry et al., 2017; Goodfellow et al., 2015; Huang et al., 2015). However, those methods need lots of extra computation to obtain adversarial examples at each time step and may tend to overfit the attacking method used in training (Buckman et al., 2018). In this paper, different from most existing methods, we tackle the problem from another perspective. In particular, we explore the possibility of designing new CNN architectures which can be trained using standard optimization methods on standard benchmark datasets but by themselves enjoy robustness, without appealing to other techniques. Recently, studies (Geirhos et al., 2017; 2018; Baker et al., 2018) show that the predictions of CNNs mainly depend on the texture of objects but not the shape. Also, Liu et al. (2018) finds attack methods usually perturb patches to contain textural features of incorrect classes. They suggest that the wrong prediction by CNNs for adversarial examples comes from the change on the texture-level information. The small perturbation of adversarial examples will change the textures and eventually affect the features extracted by the CNNs. Therefore, a natural way to avoid adversarial examples is to let the CNN make prediction

relying less on textures but more about other information which will not be severely affected by small perturbations, such as shape.

In real practice, sometimes a camera might have mechanical failures which cause the output image to have many defective pixels (such pixels are always black in all images). Nonetheless, humans can still recognize objects in the image with defective pixels but have to classify the objects by other information as some local textural information is missing. Motivated by this, we introduce the concept of defectiveness into the convolutional neural networks: We call a neuron a defective neuron if its output value is fixed to zero no matter what input signal is received, and a convolutional layer a *defective convolutional layer* if it contains defective neurons. Before training, we replace the standard convolutional layers with the defective version on a standard CNN and train the network in the standard way. As defective neurons of the defective convolutional layer contain no information and are very different from their spatial neighbors, the textural information cannot be accurately extracted from the bottom defective layers to top layers. Therefore, we destroy local textural information to a certain extent and prompt the neural network to learn more other information for classification. We call the architecture deployed with defective convolutional layers as *Defective CNN*.

We find that applying the defective convolutional layers to the bottom[1] layers of the network and introducing various patterns for defective neurons arrangement across channels are crucial for robustness. According to the experimental results, we find

- Standard CNN consistently works better than Defective CNN on manipulated images in which patches are randomly relocated. This justifies our proposal: Defective CNN makes predictions relying less on textural information but more on shape information, and thus preserving local textural information but destroying shape information of images would more hurt its performance. Furthermore, the adversarial examples generated by Defective CNNs appear to have semantic shapes (See Figure 1 and Appendix B).

- Experimental results show that Defective CNN has superior defense performance than standard CNN against the decision-based attack, transfer-based attacks, additive Gaussian noise, and grey-box attacks.

- Using the standard training method, Defective CNN achieves state-of-the-art results against two transfer-based black-box attacks while maintaining high accuracy on clean test data. This suggests that the proposed architecture may be practical for real-world tasks.

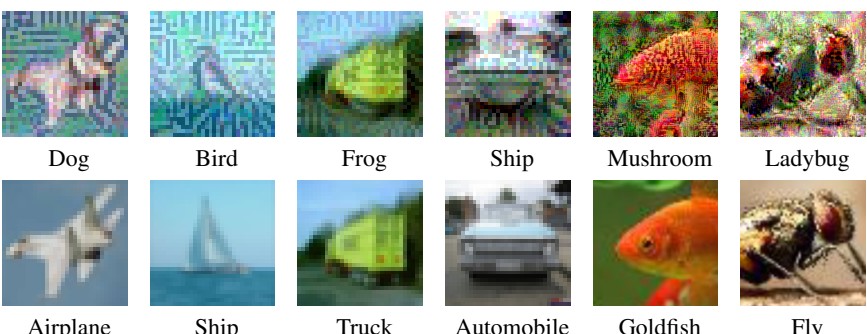

| Dog | Bird | Frog | Ship | Mushroom | Ladybug |
| Airplane | Ship | Truck | Automobile | Goldfish | Fly |

Figure 1: Adversarial examples generated by Defective CNNs. **First row**: the adversarial examples and the predicted labels. **Second row**: the corresponding original images and the ground truth labels. Detailed experimental settings and more examples can be found in Appendix B.

## 2 RELATED WORK

Various methods have been proposed to defend against adversarial examples. One line of research is to derive a meaningful optimization objective and optimize the model by adversarial training (Szegedy et al., 2013; Madry et al., 2017; Goodfellow et al., 2015; Huang et al., 2015). The high-level idea of these works is that if we can predict the potential attack to the model during optimization, then we

---

[1]In this paper, bottom layer means the layer close to the input and top layer means the layer close to the output prediction.

can give the attacked sample a correct signal and use it during training. Another line of research is to take an adjustment to the input image before letting it go through the deep neural network (Liao et al., 2017; Song et al., 2017; Samangouei et al., 2018; Sun et al., 2018). The basic intuition behind this is that if we can clean the adversarial attack to a certain extent, then such attacks can be defended. Although these methods achieve some success, a major difficulty is that it needs a large extra cost to collect adversarial examples and hard to apply on large-scale datasets.

Several studies (Geirhos et al., 2017; 2018; Baker et al., 2018) show that the prediction of CNNs is mainly from the texture of objects but not the shape. Also, Liu et al. (2018) found that adversarial examples usually perturb a patch of the original image so that the perturbed patch looks like the texture of incorrect classes. For example, the adversarial example of the panda image is misclassified as a monkey because a patch of the panda skin is perturbed adversarially so that it alone looks like the face of a monkey (see Figure 11 in Liu et al. (2018)). All previous works above suggest that the CNN learns textural information more than shape and the adversarial attack might come from textural-level perturbations. This is also correlated with robust features (Tsipras et al., 2018; Ilyas et al., 2019; Hosseini et al., 2019) which has attracted more interest recently. Pixels which encode textural information contain high redundancy and may be easily deteriorated to the distribution of incorrect classes. However, shape information is more compact and might be a more robust feature.

## 3   DEFECTIVE CONVOLUTIONAL NEURAL NETWORK

### 3.1   DESIGN OF DEFECTIVE CONVOLUTIONAL LAYERS

In this subsection, we introduce our proposed defective convolutional neural networks and discuss the differences between our proposed method and related topics.

First, we briefly introduce the notations. For one convolutional layer, denote $x$ as the input and $z$ as the output of neurons in the layer. Note that $x$ may be the input image or the output of the last convolutional layer. The input $x$ is usually a $M \times N \times K$ tensor in which $M/N$ are the height/width of a feature map, and $K$ is the number of feature maps, or equivalently, channels. Denote $w$ and $b$ as the parameters (e.g., the weights and biases) of the convolutional kernel. Then a standard convolutional layer can be mathematically defined as below.

**Standard convolutional layer:**

$$
\begin{aligned}
x' &= w \otimes_{\text{conv}} x + b, & (1)\\
z &= f(x'), & (2)
\end{aligned}
$$

where $f(\cdot)$ is a non-linear activation function such as ReLU[2] and $\otimes_{\text{conv}}$ is the convolutional operation.

The convolutional filter receives signals in a patch and extracts local textural information from the patch. As mentioned in the introduction, recent works suggest that the prediction of standard CNNs strongly depends on such textural information, and noises imposed on the texture may lead to wrong predictions. Therefore, we hope to learn a feature extractor which does not solely rely on textural features and also considers other information. To achieve this goal, we introduce the *defective convolutional layer* in which some neurons are purposely designed to be corrupted. Define $M_{\text{defect}}$ to be a binary matrix of size $M \times N \times K$. Our defective convolutional layer is defined as follows.

**Defective convolutional layer:**

$$
\begin{aligned}
x' &= w \otimes_{\text{conv}} x + b, & (3)\\
z' &= f(x') & (4)\\
z &= M_{\text{defect}} * z', & (5)
\end{aligned}
$$

where $*$ denotes element-wise product. $M_{\text{defect}}$ is a fixed matrix and is not learnable during training and testing. A simple visualization of a defective convolutional layer is shown in Figure 2. From the figure, we can see that $M_{\text{defect}}$ plays a role of "masking" out values of some neurons in the layer. This disturbs the distribution of local textural information and decouples the correlation among neurons. With the masked output $z$ as input, the feature extractor of the next convolutional layer cannot accurately capture the local textural feature from $x$. As a consequence, the textural information

---

[2]Batch normalization is popularly used on $x'$ before computing $z$. Here we simply omit this.

is hard to pass through the defective CNN from bottom to top. To produce accurate predictions, the deep neural network has to find relevant signals other than the texture, e.g., the shape. Those corrupted neurons have no severe impact on the extraction of shape information since neighbors of those neurons in the same filter are still capable of passing the shape information to the next layer.

In this paper, we find that simply setting $M_{\text{defect}}$ by random initialization is already helpful for learning a robust CNN. Before training, we sample each entry in $M_{\text{defect}}$ using Bernoulli distribution with keep probability $p$ and then fix $M_{\text{defect}}$ during training and testing. More discussions and ablation studies are provided in Section 4.

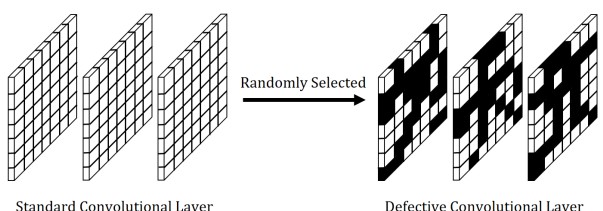

Figure 2: An illustration of three channels (neuron-wise) from standard convolutional layer to defective convolutional layer. Black neurons are the defective neurons which are randomly selected. The output of defective neurons is fixed to zero during both training and testing phases. Note that the number of parameters in the network is not reduced after including defective neurons.

As can be seen from Equation (3)-(5), the implementation of our defective convolutional layer is similar to the dropout operation (Srivastava et al., 2014). To demonstrate the relationship and differences, we mathematically define the dropout as below.

**Standard convolutional layer + dropout:**

$$
\begin{align}
M_{\text{dp}} &\sim \text{Bernoulli}(p) \tag{6}\\
x' &= w \otimes_{\text{conv}} x + b \tag{7}\\
z' &= f(x') \tag{8}\\
z &= M_{\text{dp}} * z'. \tag{9}
\end{align}
$$

The shape of $M_{\text{dp}}$ is the same as $M_{\text{defect}}$, and the value of each entry in $M_{\text{dp}}$ is sampled in each batch using some sampling strategies at each step during training. Generally, entries in $M_{\text{dp}}$ are independent and identically sampled in an online fashion using Bernoulli distribution with keep probability $p$. DropBlock (Ghiasi et al., 2018) dropouts a connected block in each channel. In SpatialDropout (Tompson et al., 2015), the dropout masks apply to whole channels. Note that the masked unit of our proposed method is a single neuron.

There are several differences between dropout and defective convolutional layer. First, the motivations behind the two methods are quite different. Dropout tries to reduce overfitting by preventing co-adaptations on training data. As the neurons in the feature maps still have full access to local textural features in testing, the model does not have to learn shape features. However, in our proposed architecture, defective neurons are fixed to be corrupted, and such neurons cannot contribute to local features. Second, the binary matrix $M_{\text{dp}}$ is sampled online during training and is removed during testing, while the binary matrix $M_{\text{defect}}$ in defective convolutional layers is predefined and keeps fixed in both training and testing. Third, places to apply and values of the keep probability $p$ for two methods are different. Dropout methods are usually applied to top layers, and $p$ is set to be large (e.g., 0.9) (Tompson et al., 2015; Ghiasi et al., 2018). For defective convolutional layer, we find using a small $p$ (e.g., 0.1) and applying it to bottom layers are more effective.

## 3.2 Defective CNN Relies Less on Texture but More on Shape

In the defective CNN, some neurons are set to be corrupted during both training and testing, and we argue that this design can help the CNN make prediction relying less on textural information but more on shape information. In this subsection, we provide some empirical analyses to verify our idea.

We design a particular image manipulation in which the local texture of the object in an image is preserved while the shape is destroyed. In detail, we divide an image into $k \times k$ patches and randomly

relocate those patches to form a new image. A typical example is shown in Figure 3. By relocating the patches, it is even hard for a human to recognize the object in the picture when $k$ is large.

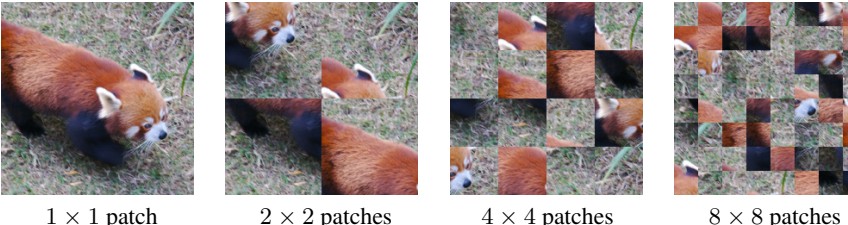

| $1 \times 1$ patch | $2 \times 2$ patches | $4 \times 4$ patches | $8 \times 8$ patches |

Figure 3: An example image that is randomly shuffled after being divided into $1 \times 1$, $2 \times 2$, $4 \times 4$ and $8 \times 8$ patches respectively.

We manipulate a set of images and test whether a defective CNN and a standard CNN can make correct predictions. The experimental details are described as follows. First, we construct a defective CNN by applying defective convolutional layers to the bottom layers of a standard ResNet-18. Then, we train the defective CNN along with a standard ResNet-18 on the ImageNet dataset and sample 4000 images from the validation set which are predicted correctly with more than 99% confidence by both two CNNs. We make manipulations to the sampled images by setting $k \in \{2, 4, 8\}$, feed these images to the networks and check their classification accuracy. The results in Table 1 show that when the shape information is destroyed but the local textural information is preserved, the defective CNN performs consistently worse than standard CNN, which verifying that defective CNN makes predictions relying less on local textural information but more on shape information.

| Model | $2 \times 2$ | $4 \times 4$ | $8 \times 8$ |
|---|---|---|---|
| Standard CNN | 99.53% | 84.36% | 20.08% |
| Defective CNN | 96.32% | 56.91% | 9.04% |

Table 1: The accuracy of standard and defective CNNs to classify randomly shuffled test images.

From another perspective, if a model predicts relying less on textural information but more on shape information, the adversarial examples against the model are likely to change the shape of the objects rather than the texture. To verify this, we train a defective CNN on CIFAR-10 dataset and Tiny-ImageNet dataset, and then apply an iterative attack method MIFGSM (Dong et al., 2017) on the validation set. Some examples are shown in Figure 1 and Appendix B. From the figures, we can see that adversarial examples against the defective CNN change the shape of the objects and thus verify our intuitions. Furthermore, additive Gaussian noises and perturbations generated by standard CNNs usually change the textural information but hard to affect the shape information. A model is supposed to defend these attacks on textures better if it predicts more relying on shape information. In Section 4, we show that defective CNNs achieve higher defense performance than standard CNN against the two types of attack, which is another evidence to support our intuitions.

## 4 EXPERIMENTS

In this section, we provide extensive experimental analyses on the performance of defective CNNs. We mainly test our models against black-box attacks. There are two reasons that indicate considering black-box attacks is meaningful. First, the black-box setting is more practical for real-world tasks. Second, for the white-box setting, the adversarial examples generated by defective CNNs appear to have semantic shapes and may even fool humans as well. This indicates that the small perturbations can actually change the semantic meaning of images for humans. Those samples should not be categorized into adversarial examples and should not be used to evaluate adversarial robustness.

For better evaluating the defense ability against black-box attacks, we propose a black-box defense evaluation protocol by examining the target model against the transfer-based attack, the decision-based attack, and the additive Gaussian noise. In real-world tasks, attackers usually cannot access the parameters of the target models and thus need to transfer adversarial examples generated by their models. This setting of attack is referred to as transfer-based attack (Liu et al., 2016). Sometimes, attackers can get the final model decision and raise the more powerful decision-based attack (Brendel

et al., 2017). Both the two types of black-box attack are available in most real-world scenarios. Recently, Ford et al. (2019) bridge the adversarial robustness and corruption robustness, and points out that a successful adversarial defend method should also effectively defense against images with additive Gaussian noise. Also, the additive Gaussian noise is a type of black-box attack to some extent since the noise distribution has nothing to do with the parameters of target models. Therefore, we also test the performance of target models against additive Gaussian noise.

We first test the robustness of defective CNN against transfer-based attacks, and then make ablation studies on possible design choices of defective CNN. Due to space limitation, more results including transfer-based attacks from ensemble models, decision-based attacks, additive Gaussian noise, white-box attacks and gray-box attacks are listed in Appendix A.

## 4.1 TRANSFER-BASED ATTACK

### 4.1.1 EXPERIMENTAL SETTINGS

We compare our proposed method with state-of-the-art defense methods (Buckman et al., 2018; Madry et al., 2017). For fair comparisons, we follow Buckman et al. (2018) to generate adversarial examples using wide residual networks (Zagoruyko & Komodakis, 2016) with a depth of 32 and a width factor of 4. The 4-block structure of ResNet-32 is shown in Appendix C. The blocks are labeled $0, 1, 2, 3$ and the $0^{\text{th}}$ block is the first convolution layer. Both FGSM (Goodfellow et al., 2015) and PGD (Kurakin et al., 2016) attacks are run on the entire validation set of CIFAR-10 dataset. These two methods both have $\ell_\infty$ perturbation scale $8$ and PGD runs for $7$ gradient descent steps with step size $2$. The generated adversarial examples are used to attack target networks. For the target network, we use the same structure but applying defective convolutional layers to the $0^{\text{th}}$ and $1^{\text{st}}$ blocks with keep probability $p = 0.1$ and train the model using standard optimization method. As is mentioned in Section 3, our proposed method is essentially different from dropout, and thus we also take dropout methods as baselines. More specifically, we test SpatialDropout and DropBlock. For both methods, we follow the instruction from Ghiasi et al. (2018) to apply dropout to the $3^{\text{rd}}$ block with $p = 0.9$. The block of DropBlock is set to be a $3 \times 3$ square. Training details can be found in Appendix D.

Second, we test our proposed method in different architectures on the CIFAR-10 dataset. We apply defective convolutional layers, in a way which is similar to the experiment above, to five popular network architectures: ResNet-18 (He et al., 2016), ResNet-50, DenseNet-121 (Huang et al., 2017), SENet-18 (Hu et al., 2017b) and VGG-19 (Simonyan & Zisserman, 2014). For each architecture, we replace the standard convolutional layer with the defective version on the bottom layers. We then test the black-box defense performance against transfer-based attacks on 5000 samples from the validation set. Adversarial examples are generated by PGD, which runs for 20 steps with step size 1 and the $\ell_\infty$ perturbation scale is set to 16. More results on the MNIST dataset and illustrations of where to apply the defective layers can be found in Appendix A and C.

### 4.1.2 EXPERIMENTAL RESULTS

First, we compare with two strong defense methods (Madry et al., 2017; Buckman et al., 2018) and two dropout methods (Tompson et al., 2015; Ghiasi et al., 2018). The results are listed in Table 2. Madry et al. (2017) is one adversarial training method that directly optimizes on adversarial examples in an online way. Based on adversarial training, Buckman et al. (2018) proposed a method that discretizes inputs and achieved higher accuracy against transfer-based attack. The results show the strengths of our proposed method on both robustness and generalization, even though our model is only trained on *clean data*. In addition, from the results of two dropout methods, we can conclude that SpatialDropout and DropBlock do not improve the robustness of standard CNNs.

| Model | FGSM | PGD | Test Accuracy |
|---|---|---|---|
| Standard CNN | 52.88% | 15.98% | 95.39% |
| Standard CNN + SD (Tompson et al., 2015) | 51.63% | 14.28% | 95.98% |
| Standard CNN + DB (Ghiasi et al., 2018) | 51.55% | 12.71% | 95.81% |
| Adversarial Training (Madry et al., 2017) | 85.60% | 86.00% | 87.30% |
| Thermometer(16) (Buckman et al., 2018) | - | 88.25% | 89.88% |
| Thermometer(32) (Buckman et al., 2018) | - | 86.60% | 90.30% |
| Defective CNN | **86.24%** | **88.43%** | **91.12%** |

Table 2: Black-box defense performances against transfer-based attacks.

Second, we list the black-box defense results of applying defective convolutional layers to various architectures in Table 3. The results show that defective convolutional layers consistently improve the robustness of various network architectures against transfer-based attacks. In this paper, the successful defense rates except Table 2 are calculated on the adversarial examples whose corresponding original images can be classified correctly by the tested model.

| Architecture | ResNet-18 | ResNet-50 | DenseNet-121 | SENet-18 | VGG-19 | Test Accuracy |
|---|---|---|---|---|---|---|
| ResNet-18 | 5.98% | 0.94% | 14.14% | 3.32% | 26.97% | 95.33% |
| 0.5-Bottom | 53.89% | 33.05% | 70.38% | 57.52% | 58.66% | 93.39% |
| 0.3-Bottom | 78.23% | 67.64% | 86.99% | 82.46% | 77.57% | 91.83% |
| ResNet-50 | 16.61% | 0.22% | 14.60% | 12.26% | 42.38% | 95.25% |
| 0.5-Bottom | 51.55% | 17.61% | 62.69% | 53.82% | 62.73% | 94.43% |
| 0.3-Bottom | 71.63% | 48.03% | 80.94% | 75.91% | 75.72% | 93.46% |
| DenseNet-121 | 14.53% | 0.60% | 2.98% | 7.79% | 31.57% | 95.53% |
| 0.5-Bottom | 35.07% | 8.01% | 34.21% | 30.86% | 45.28% | 94.34% |
| 0.3-Bottom | 58.19% | 33.86% | 62.32% | 59.74% | 62.09% | 92.82% |
| SENet-18 | 6.72% | 0.90% | 12.29% | 2.23% | 26.86% | 95.09% |
| 0.5-Bottom | 52.95% | 30.78% | 66.81% | 52.49% | 57.45% | 93.53% |
| 0.3-Bottom | 74.73% | 59.42% | 84.31% | 78.72% | 75.04% | 92.54% |
| VGG-19 | 33.46% | 14.16% | 49.76% | 29.98% | 21.20% | 93.93% |
| 0.5-Bottom | 72.27% | 59.70% | 83.50% | 77.93% | 66.75% | 91.73% |
| 0.3-Bottom | 85.53% | 79.20% | 91.01% | 88.51% | 81.92% | 90.11% |

Table 3: Black-box defense performances against transfer-based attacks. Numbers in the middle mean the success defense rates. Networks in the first row are the source models for generating adversarial examples by PGD. 0.5-Bottom and 0.3-Bottom mean applying defective convolutional layers with keep probability 0.5 and 0.3 to the bottom layers of the network whose name lies just above them. The source and target networks are initialized differently if they share the same architecture.

## 4.2 ABLATION STUDIES

There are several design choices of the defective CNN, which include the appropriate positions to apply defective convolutional layers, the benefit of breaking symmetry, the diversity introduced by randomness, as well as the extensibility of defective layers via structure adjustment. In this subsection, we conduct a series of comparative experiments and use black-box defense performance against transfer-based attacks as the evaluation criterion. In our experiments, we found that the performance is not sensitive to the choices on the source model to attack and the target model to defense. Without loss of generality, we only list the performances using DenseNet-121 as the source model and ResNet-18 as the target model on the CIFAR-10 dataset and leave more experimental results in Appendix A.8. The results are listed in Table 4.

**Defective Layers on Bottom layers v.s Top Layers.** We apply defective layers with different keep probabilities to the bottom layers and the top layers of the original CNN, respectively. Comparing the results of the models with the same keep probability but different parts being masked, we find that applying defective layers to *bottom* layers enjoys significantly higher success defense rates. Moreover, only applying defective layers to bottom layers can achieve better performance than applying defective layers on both the bottom and top layers. The bottom layers mainly contribute to detect the edges and shape, while the receptive fields of neurons in top layers are too large to respond to the location sensitive information. This corroborates the phenomena shown in Zeiler & Fergus (2014); Mordvintsev et al. (2015). Also, we find that the defense accuracy monotonically increases as the test accuracy decreases along with the keep probability (See the trend map in Appendix A.1). The appropriate value for the keep probability mainly depends on the relative importance of generalization and robustness.

**Defective Neuron v.s. Defective Channel.** As our method independently selects defective neurons on different channels in a layer, we break the symmetry of the original CNN structure. To see whether this asymmetric structure would help, we try to directly mask whole channels instead of neurons using the same keep probability as the defective layer and train it to see the performance. This defective channel method does not hurt the symmetry while also leading to the same decrease in the number of convolutional operations. Table 4 shows that although our defective CNN suffers a

| Architecture | FGSM$_{16}$ | PGD$_{16}$ | PGD$_{32}$ | CW$_{40}$ | Test Accuracy |
|---|---|---|---|---|---|
| ResNet-18 | 14.91% | 14.14% | 7.16% | 8.23% | 95.33% |
| 0.7-Bottom | 23.29% | 51.29% | 37.00% | 36.95% | 94.03% |
| 0.5-Bottom | 30.86% | 70.38% | 56.36% | 54.02% | 93.39% |
| 0.3-Bottom | 48.57% | 86.99% | 78.41% | 73.70% | 91.83% |
| 0.7-Top | 14.62% | 10.55% | 4.91% | 7.88% | 95.16% |
| 0.5-Top | 10.76% | 11.06% | 5.10% | 7.19% | 94.94% |
| 0.3-Top | 11.23% | 11.77% | 5.80% | 10.10% | 94.61% |
| 0.7-Bottom, 0.7-Top | 24.15% | 45.12% | 30.24% | 29.65% | 94.16% |
| 0.7-Bottom, 0.3-Top | 11.26% | 33.67% | 20.28% | 23.31% | 93.44% |
| 0.3-Bottom, 0.3-Top | 27.43% | 75.49% | 62.78% | 62.47% | 89.78% |
| 0.3-Bottom, 0.7-Top | 40.58% | 82.77% | 72.15% | 68.58% | 91.23% |
| 0.5-Bottom | 30.86% | 70.38% | 56.36% | 54.02% | 93.39% |
| 0.1-Bottom | 79.93% | 96.70% | 94.68% | 89.67% | 87.68% |
| 0.5-Bottom$_{DC}$ | 12.15% | 19.93% | 11.20% | 12.72% | 95.12% |
| 0.1-Bottom$_{DC}$ | 19.00% | 53.87% | 41.40% | 44.80% | 93.27% |
| 0.5-Bottom$_{SM}$ | 48.86% | 85.00% | 75.60% | 72.07% | 92.57% |
| 0.1-Bottom$_{SM}$ | 39.40% | 80.36% | 72.43% | 65.38% | 74.28% |
| 0.5-Bottom$_{\times 2}$ | 20.78% | 50.16% | 34.24% | 34.00% | 94.12% |
| 0.1-Bottom$_{\times 2}$ | 68.83% | 93.35% | 88.25% | 82.74% | 90.49% |

Table 4: Ablation experiments of defective CNN. Numbers in the middle mean the success defense rates. $p$-**Bottom** and $p$-**Top** mean applying defective layers with keep probability $p$ to bottom layers and top layers respectively. $p$-**Bottom**$_{DC}$ means making whole channels defective with keep probability $p$. $p$-**Bottom**$_{SM}$ means using the same defective mask in every channel with keep probability $p$. $p$-**Bottom**$_{\times n}$ means increasing channel number to $n$ times at defective layers. FGSM$_{16}$, PGD$_{16}$ and PGD$_{32}$ denote attack method FGSM with $\ell_\infty$ perturbation scale 16, PGD with $\ell_\infty$ perturbation scale 16 and 32 respectively. CW$_{40}$ denotes CW attack method (Carlini & Wagner, 2016) with confidence $\kappa = 40$.

small drop in test accuracy due to the low keep probability, we have a great gain in the robustness, compared with the defective-channel CNN.

**Defective Masks are Shared Among Channels or Not.** The randomness in generating masks in different channels and layers allows each convolutional filter to focus on different input patterns. Also, it naturally involves various topological structures for local feature extraction instead of learning (Dai et al., 2017; Chang et al., 2018). We show the essentiality of generating various masks per layer via experiments that compare to a method that only randomly generates one mask per layer and uses it in every channel. Table 4 shows that applying the same mask to each channel will decrease the test accuracy. This may result from the limitation of expressivity due to the monotone masks at every channel of the defective layer.

**Increase the Number of Channels at Defective Layers.** Although masking neurons does not reduce the parameters of the CNN, it reduces the number of convolutional operations and may decrease the expressive capacity of the CNN. To compensate for these defective positions, we increase the number of neurons at the defective layers by increasing the number of channels. Table 4 shows that increasing channels does help the network with defective layers to obtain higher test accuracy while maintaining good robustness performance.

## 5    CONCLUSION

In this paper, we introduce and experiment on defective CNNs, a modified version of existing CNNs that makes CNNs capture more information other than local textures, especially the shape. We show that defective CNNs can achieve much better robustness while maintaining high test accuracy. More specifically, by using defective convolutional layers, we reach state-of-the-art performance against two transfer-based attack methods. Another insight resulting from our experiments is that the adversarial perturbations generated against defective CNNs can actually change the semantic information of images and may even fool humans. We hope that these findings bring more understanding on adversarial examples and the robustness of neural networks.

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

## A    MORE EXPERIMENTAL RESULTS

### A.1    BLACK-BOX ATTACK WITH DIFFERENT KEEP PROBABILITIES

In this subsection, we show the trade-off between robustness and generalization performance in defective CNNs with different keep probabilities. We use DenseNet-121 (Huang et al., 2017) as the source model to generate adversarial examples from CIFAR-10 with PGD (Kurakin et al., 2016), which runs for 20 steps with step size 1 and perturbation scale 16. The defective convolutional layers are applied to the bottom layers of ResNet-18 (He et al., 2016). Figure 4 shows that the defense accuracy monotonically increases as the test accuracy decreases along with the keep probability. We can see the trade-off between robustness and generalization.

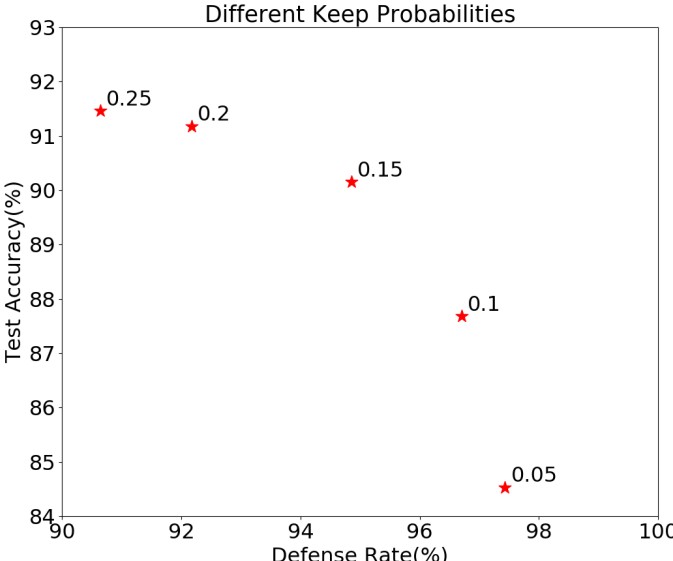

Figure 4: Relationship between success defense rates against adversarial examples generated by PGD and test accuracy with respect to different keep probabilities. Each red star represents a specific keep probability with its value written near the star.

## A.2 TRANSFER-BASED ATTACK FROM ENSEMBLE MODELS ON CIFAR-10

In this subsection, we evaluate the defense performance of networks with defective convolutional layers against transfer-based attack from ensemble models on the CIFAR-10 dataset. We apply defective convolutional layers to five popular network architectures ResNet-18, ResNet-50 (He et al., 2016), DenseNet-121, SENet-18 (Hu et al., 2017a), VGG-19 (Simonyan & Zisserman, 2014), and test the black-box defense performance against transfer-based attacks from ensemble models on the CIFAR-10 dataset. For each architecture, we replace the standard convolutional layer with the defective version on the bottom layers of different architectures. Illustrations of defective layers applied to these network architectures can be found in Appendix C. We test the black-box defense performance against transfer-based attacks on 5000 samples from the validation set. Adversarial examples are generated by PGD, which runs for 7 steps with step size 2 and the $\ell_\infty$ perturbation scale is set to 8. We generate five ensemble models as the source model by fusing every four models in all five models.

The results can be found in Table 5. These results show that defective convolutional layers can consistently improve the black-box defense performance of various network architectures against transfer-based attacks from ensemble models on the CIFAR-10 dataset.

| Architecture | −ResNet-18 | −ResNet-50 | −DenseNet-121 | −SENet-18 | −VGG-19 | Test Accuracy |
|---|---|---|---|---|---|---|
| ResNet-18 | 1.02% | 0.74% | 0.76% | 0.94% | 0.88% | 95.33% |
| 0.5-Bottom | 32.98% | 35.95% | 29.36% | 32.31% | 37.24% | 93.39% |
| 0.3-Bottom | 69.52% | 72.44% | 67.02% | 68.63% | 72.23% | 91.83% |
| ResNet-50 | 1.07% | 2.32% | 1.31% | 1.17% | 0.82% | 95.25% |
| 0.5-Bottom | 23.61% | 31.52% | 21.20% | 22.89% | 25.59% | 94.43% |
| 0.3-Bottom | 55.47% | 62.43% | 53.25% | 55.13% | 58.47% | 93.46% |
| DenseNet-121 | 0.70% | 0.88% | 1.37% | 0.74% | 0.58% | 95.53% |
| 0.5-Bottom | 6.99% | 10.19% | 7.77% | 6.93% | 8.10% | 94.34% |
| 0.3-Bottom | 32.07% | 38.28% | 31.59% | 31.01% | 35.42% | 92.82% |
| SENet-18 | 0.91% | 0.66% | 0.74% | 0.92% | 0.64% | 95.09% |
| 0.5-Bottom | 29.16% | 32.58% | 25.84% | 29.17% | 32.88% | 93.53% |
| 0.3-Bottom | 62.12% | 65.83% | 59.94% | 62.90% | 65.64% | 92.54% |
| VGG-19 | 8.27% | 8.28% | 6.42% | 7.64% | 14.08% | 93.93% |
| 0.5-Bottom | 60.62% | 63.71% | 57.85% | 59.28% | 64.73% | 91.73% |
| 0.3-Bottom | 80.97% | 82.84% | 80.06% | 80.04% | 82.86% | 90.11% |

Table 5: Black-box defense performances against transfer-based attacks from ensemble models on the CIFAR-10 dataset. Numbers in the middle mean the success defense rates. Networks in the first row indicate the source models which ensemble other four models except for the network itself. The source model generates adversarial examples by PGD. 0.5-Bottom and 0.3-Bottom mean applying defective convolutional layers with keep probability 0.5 and 0.3 to the bottom layers of the network whose name lies just above them. The source and target networks are initialized differently if they share the same architecture.

## A.3 TRANSFER-BASED ATTACK ON MNIST

In this subsection, we evaluate the defense performance of networks with defective convolutional layers against trasfer-based attack on the MINST dataset. We apply defective convolutional layers to five popular network architectures ResNet-18, ResNet-50, DenseNet-121, SENet-18, VGG-19, and test the black-box defense performance against transfer-based attacks on MNIST dataset. For each architecture, we replace the standard convolutional layer with the defective version on bottom layers of different architectures. Illustrations of defective layers applied to these network architectures can be found in Appendix C. We test the black-box defense performance against transfer-based attacks on 5000 samples from the validation set. Adversarial examples are generated by PGD which runs for 40 steps with step size $0.01 \times 255$ and perturbation scale $0.3 \times 255$.

The results can be found in Table 6. These results show that defective convolutional layers can consistently improve the black-box defense performance of various network architectures against transfer-based attacks on the MNIST dataset.

| Architecture | ResNet-18 | ResNet-50 | DenseNet-121 | SENet-18 | VGG-19 | Test Accuracy |
|---|---|---|---|---|---|---|
| ResNet-18 | 0.06% | 24.13% | 1.66% | 0.14% | 9.57% | 99.49% |
| 0.5-Bottom | 3.49% | 43.04% | 8.66% | 7.66% | 26.47% | 99.34% |
| 0.3-Bottom | 25.91% | 75.59% | 36.19% | 38.03% | 64.63% | 99.29% |
| ResNet-50 | 2.93% | 9.30% | 7.68% | 5.94% | 19.68% | 99.39% |
| 0.5-Bottom | 8.54% | 23.06% | 8.76% | 10.09% | 28.49% | 99.32% |
| 0.3-Bottom | 10.91% | 36.44% | 16.04% | 14.55% | 39.57% | 99.28% |
| DenseNet-121 | 0.48% | 29.81% | 0.02% | 1.64% | 9.90% | 99.48% |
| 0.5-Bottom | 2.57% | 35.85% | 1.10% | 3.93% | 16.29% | 99.46% |
| 0.3-Bottom | 7.13% | 58.92% | 3.37% | 11.39% | 32.69% | 99.38% |
| SENet-18 | 0.22% | 18.77% | 2.34% | 0.10% | 13.75% | 99.41% |
| 0.5-Bottom | 3.37% | 24.09% | 6.90% | 2.21% | 17.24% | 99.35% |
| 0.3-Bottom | 11.97% | 51.63% | 14.39% | 16.45% | 40.23% | 99.31% |
| VGG-19 | 3.83% | 51.77% | 5.59% | 7.34% | 3.25% | 99.48% |
| 0.5-Bottom | 12.47% | 61.18% | 12.91% | 21.71% | 19.32% | 99.37% |
| 0.3-Bottom | 29.14% | 70.59% | 31.55% | 41.87% | 47.65% | 99.33% |

Table 6: Black-box defense performances against transfer-based attacks on the MNIST dataset. Numbers in the middle mean the success defense rates. Networks in the first row are the source models for generating adversarial examples by PGD. 0.5-Bottom and 0.3-Bottom mean applying defective convolutional layers with keep probability 0.5 and 0.3 to the bottom layers of the network whose name lies just above them. The source and target networks are initialized differently if they share the same architecture.

## A.4 DECISION-BASED ATTACK

In this subsection, we evaluate the defense performance of networks with defective convolutional layers against the decision-based attack. Decision-based attack performs based on the prediction of the model. It needs less information from the model and has the potential to perform better against adversarial defenses based on gradient masking. Boundary attack (Brendel et al., 2017) is one effective decision-based attack. The attack will start from a point that is already adversarial by applying a large scale perturbation to the original image and keep decreasing the distance between the original image and the adversarial example by random walks. After iterations, we will get the final perturbation, which has a relatively small scale. The more robust the model is, the larger the final perturbation will be.

In our experiments, we use the implementation of boundary attack in Foolbox (Rauber et al., 2017). It finds the adversarial initialization by simply adding large scale uniform noises on input images. We perform our method on ResNet-18 and test the performance on CIFAR-10 with 500 samples from the validation set. The 5-block structure of ResNet-18 is shown in Appendix Figure 2. The blocks are labeled $0, 1, 2, 3, 4$ and the $0^{\text{th}}$ block is the first convolution layer. We apply the defective layer structure with keep probability $p = 0.1$ to the bottom blocks (the $0^{\text{th}}, 1^{\text{st}}, 2^{\text{nd}}$ blocks). For comparison, we implement label smoothing (Szegedy et al., 2016) with smoothing parameter $\epsilon = 0.1$ on a standard ResNet-18. We compare with both standard CNN and label smoothing (Szegedy et al., 2016) which is known to be a gradient masking method (Papernot et al., 2016a). The median squared $\ell_2$-distance of final perturbation across all samples proposed in Brendel et al. (2017) is used as our evaluation criterion. The score $S(M)$ is defined in Equation (10), where $P_i^M \in \mathbb{R}^N$ is the final perturbation that the Boundary attack finds on model $M$ for the $i^{\text{th}}$ image. Before computing $P_i^M$, the images are normalized into $[0, 1]^N$.

$$S(M) = \underset{i}{\text{Median}} \left( \frac{1}{N} \| P_i^M \|_2^2 \right) \qquad (10)$$

From the results in Table 7, we point out that the gradient masking method does not increase the robustness against boundary attack. Our proposed method achieves significant improvement over the standard CNN.

| Model | $S(M)$ |
|---|---|
| Standard CNN | 7.3e-06 |
| Standard CNN + LS | 6.8e-06 |
| Defective CNN | **3.5e-05** |

Table 7: Black-box defense performances against decision-based attack (See Section A.4 for the complete setting). The larger value $S(M)$ (defined in Equation (10)) has, the more robust the model is. LS means label smoothing.

## A.5 ADDITIVE GAUSSIAN NOISE

In this subsection, we evaluate the defense performance of networks with defective convolutional layers against additive Gaussian noise. Recently, Ford et al. (2019) bridge the adversarial robustness and corruption robustness, and points out that a successful adversarial defense method should also effectively defense against images with additive Gaussian noise. Also the Gaussian noises usually do not change the shape of objects, our models should have better defense performance. To see whether our structure is more robust in this setting, we feed input images with additive Gaussian noises to both standard and defective CNNs.

To obtain noises of scales similar to the adversarial perturbations, we generate i.i.d. Gaussian random variables $x \sim N(0, \sigma^2)$, where $\sigma \in \{1, 2, 4, 8, 12, 16, 20, 24, 28, 32\}$, clip them to the range $[-2\sigma, 2\sigma]$ and then add them to every pixel of the input image. For CIFAR-10, we add Gaussian noises to 5000 samples which are drawn randomly from the validation set and can be classified correctly by all the tested models. We place the defective layers with keep probability $p = 0.1$ on ResNet-18 in the same way as we did in Section A.4.

The experimental results are shown in Figure 5. Standard ResNet-18 is still robust to small scale Gaussian noise such as $\sigma \leq 8$. After that, the performance of standard ResNet-18 begins to drop sharply as $\sigma$ increases. In contrast, defective CNNs show far better robustness than the standard version. The defective ResNet-18 with keep probability 0.1 can maintain high accuracy until $\sigma$ increase to 16 and have a much slower downward trend as $\sigma$ increases.

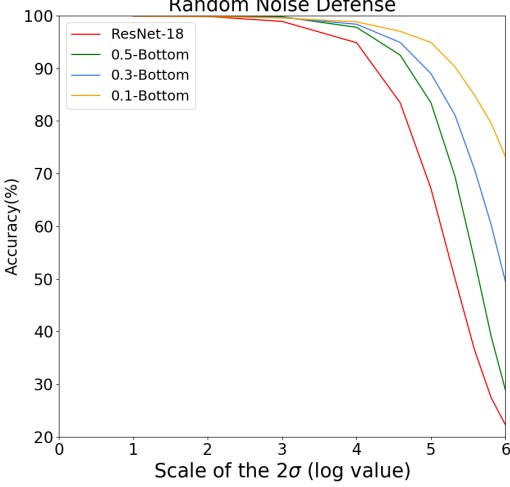

Figure 5: Defense performance against images with additive Gaussian noise, Defective versus Standard CNN. $p$-Bottom means applying defective convolutional layers with keep probability $p$ to the bottom layers of ResNet-18.

## A.6 WHITE-BOX ATTACK

In this subsection, we evaluate the defense performance of ResNet-18 with defective convolutional layers against white-box attacks on CIFAR-10 dataset. With neither obfuscated gradient nor gradient

masking, defective convolutional layers can still improve defense performance under various kinds of white-box attack (See Table 8). The results on other network architectures are similar.

| Architecture | $FGSM_1$ | $FGSM_2$ | $FGSM_4$ | $PGD_2$ | $PGD_4$ | $PGD_8$ | Test Accuracy |
|---|---|---|---|---|---|---|---|
| ResNet-18 | 81.24% | 65.78% | 51.24% | 23.80% | 3.16% | 0.02% | 95.33% |
| 0.5-Bottom | 85.22% | 68.65% | 52.04% | 38.16% | 6.67% | 0.12% | 93.39% |
| 0.3-Bottom | 85.70% | 69.69% | 54.51% | 49.01% | 18.93% | 2.86% | 91.83% |

Table 8: Defense performances against white-box attacks. Numbers in the middle mean the success defense rates. $FGSM_1, FGSM_2, FGSM_4$ refer to FGSM with perturbation scale 1,2,4 respectively. $PGD_2, PGD_4, PGD_8$ refer to PGD with perturbation scale 2,4,8 and step number 4,6,10 respectively. The step size of all PGD methods are set to 1.

## A.7 Gray-box Attack

In this subsection, we show the gray-box defense performance of defective CNNs on the CIFAR-10 dataset. We use gray-box attacks in the following two ways. One way is to generate adversarial examples against one trained neural network and test those images on a network with the same structure but different initializations. The other way is specific to our defective models. We generate adversarial examples on one trained defective CNN and test them on a network with the same keep probability but different sampling of defective neurons. In both of these two ways, the adversarial knows some information on the structure of the network but does not know the specific parameters of it.

| Architecture | 0.5-Bottom | 0.3-Bottom |
|---|---|---|
| 0.5-Bottom | 30.90% | 40.49% |
| 0.5-Bottom$_{DIF}$ | 32.77% | 40.39% |
| 0.3-Bottom | 59.24% | 36.84% |
| 0.3-Bottom$_{DIF}$ | 57.45% | 37.04% |

Table 9: Defense performances against two kinds of gray-box attacks for defective CNNs. Numbers mean the success defense rates. Networks in the first row are the source models for generating adversarial examples by PGD, which runs for 20 steps with step size 1 and perturbation scale $\ell_\infty = 16$. 0.5-Bottom and 0.3-Bottom in the left column represent the networks with the same structure as the corresponding source networks but with different initialization. 0.5-Bottom$_{DIF}$ and 0.3-Bottom$_{DIF}$ in the left column represent the networks with the same keep probabilities as the corresponding source networks but with different sampling of defective neurons.

From the results listed in Table 9, we find that defective CNNs have similar performance on adversarial examples generated by our two kinds of gray-box attacks. This phenomenon indicates that defective CNNs with the same keep probability would catch similar information which is insensitive to the selections of defective neurons. Also, comparing with the gray-box performance of standard CNNs (See Table 10), defective CNNs show stronger defense ability.

| Architecture | ResNet-18 | DenseNet-121 |
|---|---|---|
| ResNet-18 | 5.98% | 14.14% |
| DenseNet-121 | 14.53% | 2.98% |

Table 10: Defense performances against gray-box attacks for standard CNNs. Numbers mean the success defense rates. Networks in the first row are the source models for generating adversarial examples by PGD, which runs for 20 steps with step size 1 and perturbation scale $\ell_\infty = 16$. The diagonal shows gray-box performances in the setting that the source and target networks share the same structure but with different initializations.

## A.8 FULL INFORMATION ON EXPERIMENTS MENTIONED IN SECTION 4.3

In this subsection, we will show more experimental results on defective CNNs using different adversarial examples, different attack methods and different mask settings on ResNet-18. The networks used to generate adversarial examples including ResNet-18, ResNet-50, DenseNet-121, SENet18, and VGG-19. More specifically, we choose 5000 samples to generate adversarial examples via FGSM and PGD, and 1000 samples for CW attack. All samples are drawn from the validation set of CIFAR-10 dataset and can be correctly classified correctly by the model used to generate adversarial examples.

For FGSM, we try step size $\epsilon \in \{8, 16, 32\}$, namely **FGSM**$_8$, **FGSM**$_{16}$, **FGSM**$_{32}$, to generate adversarial examples. For PGD, we have tried more extensive settings. Let $\{\epsilon, T, \alpha\}$ be the PGD setting with step size $\epsilon$, the number of steps $T$ and the perturbation scale $\alpha$, then we have tried PGD settings $(1, 8, 4), (2, 4, 4), (4, 2, 4), (1, 12, 8), (2, 6, 8), (4, 3, 8), (1, 20, 16), (2, 10, 16), (4, 5, 16), (1, 40, 32),$ $(2, 20, 32), (4, 10, 32)$ to generate PGD adversarial examples. From the experimental results, we observe the following phenomena. First, we find that the larger the perturbation scale is, the stronger the adversarial examples are. Second, for a fixed perturbation scale, the smaller the step size is, the more successful the attack is, as it searches the adversarial examples in a more careful way around the original image. Based on these observation, we only show strong PGD attack results in the Appendix, namely the settings $(1, 20, 16)$ (**PGD**$_{16}$), $(2, 10, 16)$ (**PGD**$_{2,16}$) and $(1, 40, 32)$ (**PGD**$_{32}$). Nonetheless, our models also perform much better on weak PGD attacks. For the CW attack, we have also tried different confidence parameters $\kappa$. However, we find that for large $\kappa$, the algorithm is hard to find adversarial examples for some neural networks such as VGG because of its logit scale. For smaller $\kappa$, the adversarial examples have weak transferability, which means they can be easily defended even by standard CNNs. Therefore, in order to balance these two factors, we choose $\kappa = 40$ (**CW**$_{40}$) for DenseNet-121, ResNet-50, SENet-18 and $\kappa = 20$ (**CW**$_{20}$) for ResNet-18 as a good choice to compare our models with standard ones. The step number for choosing the parameter $c$ is set to 30.

Note that the noises of FGSM and PGD are considered in the sense of $\ell_\infty$ norm and the noise of CW is considered in the sense of $\ell_2$ norm. All adversarial examples used to evaluate can fool the original network. Table 11,12,13,14 and 15 list our experimental results. DC means we replace defective neurons with defective channels in the corresponding blocks to achieve the same keep probability. SM means we use the same defective mask on all the channels in a layer. $\times n$ means we multiply the number of the channels in the defective blocks by $n$ times. EN means we ensemble five models with different defective masks of the same keep probability.

## B ADVERSARIAL EXAMPLES GENERATED BY DEFECTIVE CNNS

### B.1 ADVERSARIAL EXAMPLES THAT CAN FOOL HUMAN

In this subsection, we show more adversarial examples generated by defective CNNs. Figure 6 shows some adversarial examples generated on the CIFAR-10 dataset along with the corresponding original images. These examples are generated from CIFAR-10 against a defective ResNet-18 of keep probability 0.2 on the $0^{\text{th}}$, $1^{\text{st}}$, $2^{\text{nd}}$ blocks, a defective ResNet-18 of keep probability 0.1 on the $1^{\text{st}}$, $2^{\text{nd}}$ blocks, and a standard ResNet-18. We use attack method MIFGSM with perturbation scale $\alpha = 16$ and $\alpha = 32$. We also show some adversarial examples generated from Tiny-ImageNet[3] along with the corresponding original images in Figure 7. These examples are generated from Tiny-ImageNet against a defective ResNet-18 of keep probability of the keep probability 0.1 on the $1^{\text{st}}$, $2^{\text{nd}}$ blocks and a standard ResNet-18. The attack methods are MIFGSM with scale 64 and 32, step size 1 and step number 40 and 80 respectively.

The adversarial examples generated by defective CNNs exhibit more semantic shapes of their fooled classes, such as the mouth of the frog in Figure 6. This also corroborates the point made in Tsipras et al. (2018) that more robust models will be more aligned with human perception.

---

[3]https://tiny-imagenet.herokuapp.com/

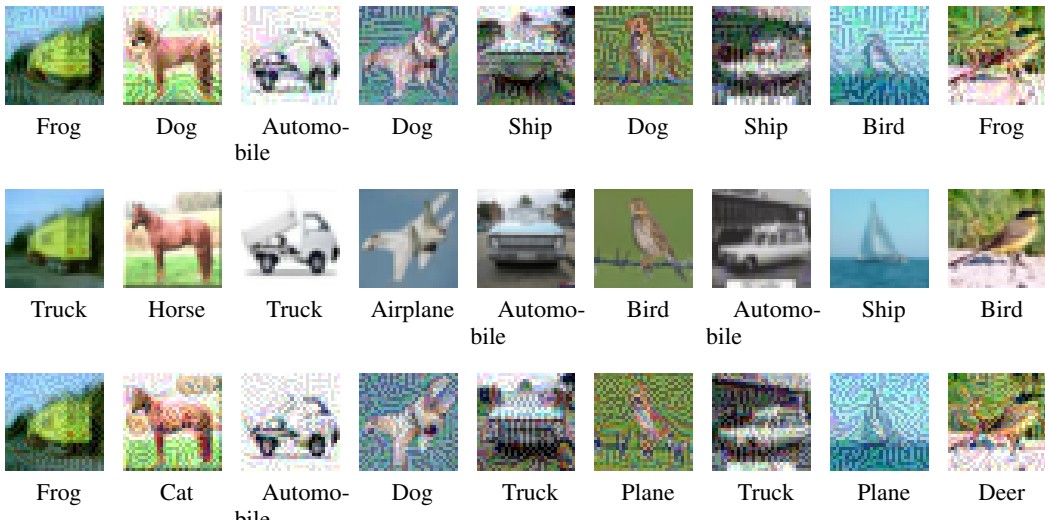

Figure 6: CIFAR-10 dataset. **First row**: the adversarial examples generated by defective CNNs and the predicted labels. **Second row**: original images. **Second row**: the adversarial examples generated by the standard CNN and the predicted labels.

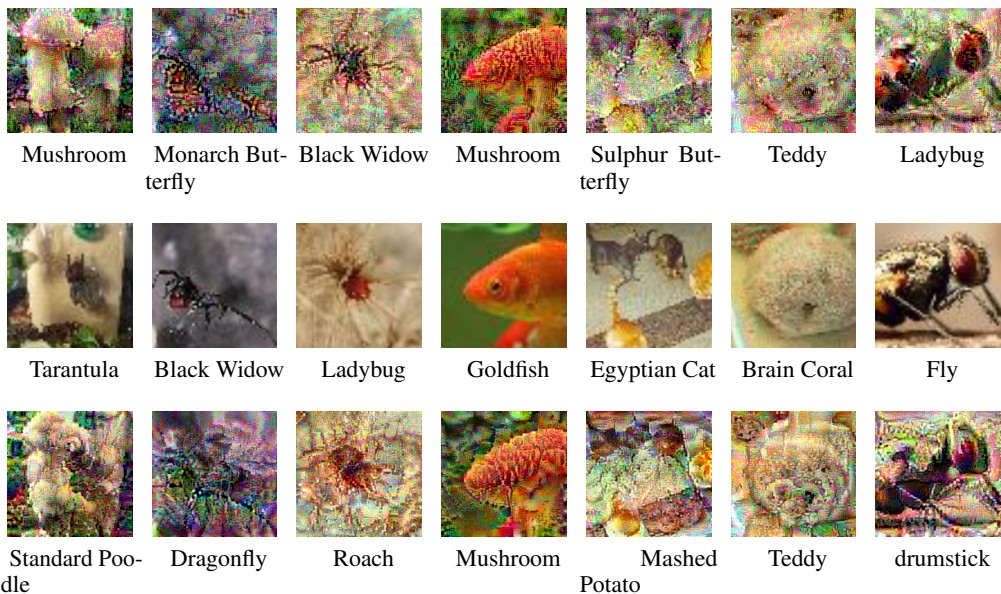

Figure 7: Tiny-ImageNet dataset. **First row**: the adversarial examples generated by defective CNNs and the predicted labels. **Second row**: original images. **Second row**: the adversarial examples generated by the standard CNN and the predicted labels.

## B.2 RANDOMLY SELECTED ADVERSARIAL EXAMPLES

See Figure 15 for a randomly sampled set of images from Tiny-ImageNet along with the corresponding adversarial examples generated against standard and defective ResNet-18 under the same attack setting.

## C   ARCHITECTURES

In this subsection, we briefly introduce the network architectures used in our experiments. Generally, we apply defective convolutional layers to the bottom layers of the networks and we have tried six different architectures, namely **ResNet-18**, **ResNet-50**, **DenseNet-121**, **SENet-18**, **VGG-19** and **WideResNet-32**. We next illustrate these architectures and show how we apply defective convolutional layers to them. In our experiments, applying defective convolutional layers to a block means randomly selecting defective neurons in every layer of the block.

### C.1   RESNET-18

ResNet-18 (He et al., 2016) contains $5$ blocks: the $0^{th}$ block is one single $3 \times 3$ convolutional layer, and each of the rest contains four $3 \times 3$ convolutional layers. Figure 8 shows the whole structure of ResNet-18. In our experiments, we apply defective convolutional layers to the $0^{th}, 1^{st}, 2^{nd}$ blocks which are the bottom layers.

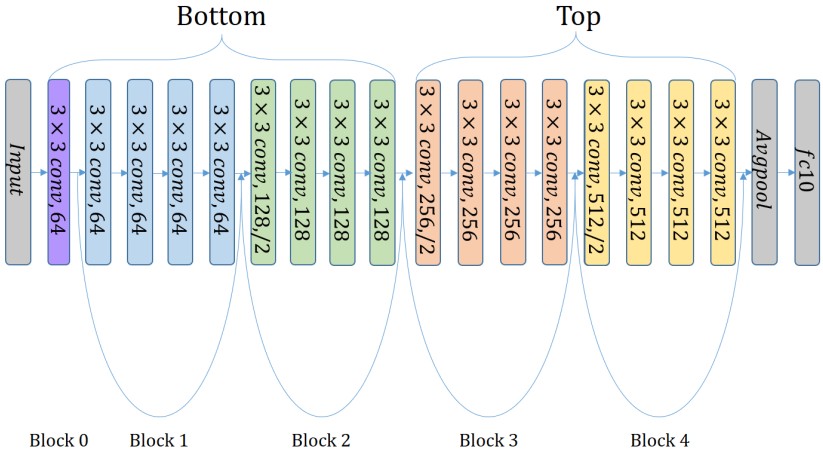

Figure 8: The architecture of ResNet-18

### C.2   RESNET-50

Similar to ResNet-18, ResNet-50 (He et al., 2016) contains $5$ blocks and each block contains several $1 \times 1$ and $3 \times 3$ convolutional layers (i.e. Bottlenecks). In our experiment, we apply defective convolutional layers to the $3 \times 3$ convolutional layers in the first three bottom blocks. The defective layers in the $1^{st}$ block are marked by the red arrows in Figure 9.

### C.3   DENSENET-121

DenseNet-121 (Huang et al., 2017) is another popular network architecture in deep learning researches. Figure 10 shows the whole structure of DenseNet-121. It contains $5$ Dense-Blocks, each of which contains several $1 \times 1$ and $3 \times 3$ convolutional layers. Similar to what we do for ResNet-50, we apply defective convolutional layers to the $3 \times 3$ convolutional layers in the first three "bottom" blocks. The growth rate is set to $32$ in our experiments.

### C.4   SENET-18

SENet (Hu et al., 2017a), a network architecture which won the first place in ImageNet contest 2017, is shown in Figure 11. Note that here we use the pre-activation shortcut version of SENet-18 and we apply defective convolutional layers to the convolutional layers in the first 3 SE-blocks.

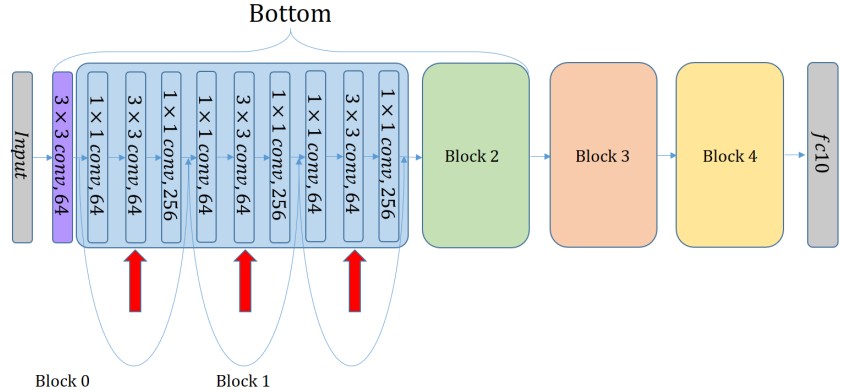

Figure 9: The architecture of ResNet-50

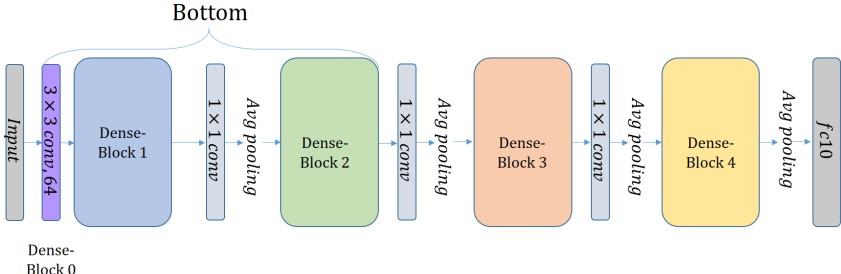

Figure 10: The architecture of DenseNet-121

### C.5    VGG-19

VGG-19 (Simonyan & Zisserman, 2014) is a typical neural network architecture with sixteen $3 \times 3$ convolutional layers and three fully-connected layers. We slightly modified the architecture by replacing the final 3 fully connected layers with 1 fully connected layer as is suggested by recent architectures. Figure 12 shows the whole structure of VGG-19. We apply defective convolutional layers on the first four $3 \times 3$ convolutional layers.

### C.6    WIDERESNET-32

Based on residual networks, Zagoruyko & Komodakis (2016) proposed a wide version of residual networks which have much more channels. In our experiments, we adopt the network with a width factor of 4 and apply defective layers on the 0th and 1st blocks. Figure 13 shows the whole structure of WideResNet-32.

### D    TRAINING PROCESS ON CIFAR-10 AND MNIST

To guarantee our experiments are reproducible, here we present more details on the training process in our experiments. When training models on CIFAR-10, we first subtract per-pixel mean. Then we apply a zero-padding of width 4, a random horizontal flip and a random crop of size $32 \times 32$ on train data. No other data augmentation method is used. We apply SGD with momentum parameter $0.9$, weight decay parameter $5 \times 10^{-4}$ and mini-batch size 128 to train on the data for 350 epochs. The learning rate starts from $0.1$ and is divided by 10 when the number of epochs reaches 150 and 250. When training models on MNIST, we first subtract per-pixel mean. Then we apply random horizontal flip on train data. We apply SGD with momentum parameter $0.9$, weight decay parameter $5 \times 10^{-4}$ and mini-batch size 128 to train on the data for 50 epochs. The learning rate starts from $0.1$ and is divided by 10 when the number of epochs reaches 20 and 40. Figure 14 shows the train and test

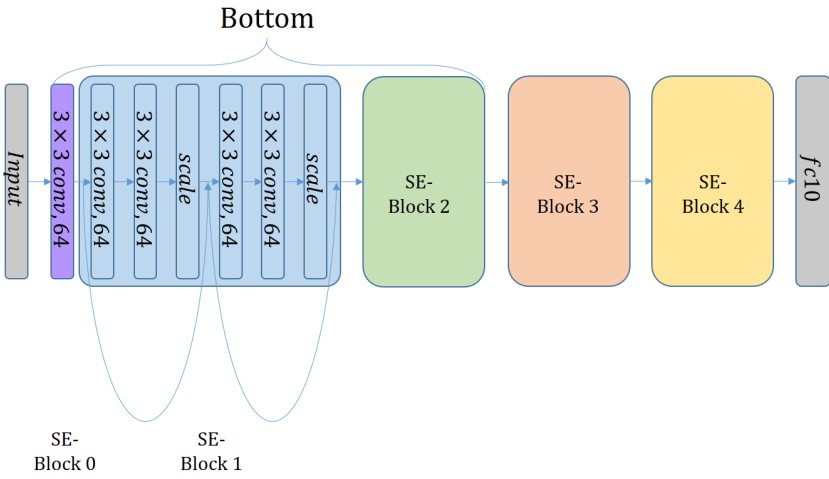

Figure 11: The architecture of SENet-18

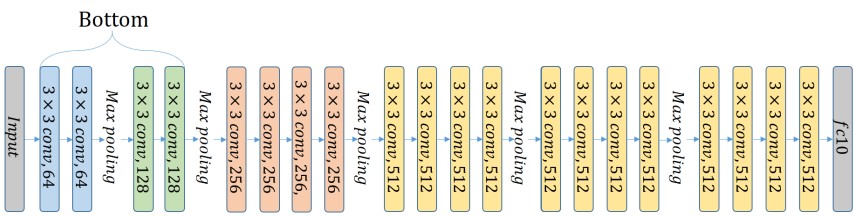

Figure 12: The architecture of VGG-19

curves of standard and defective ResNet-18 on CIFAR-10 and MNIST. Different network structures share similar tendency regarding the train and test curves.

## E ATTACK APPROACHES

In this subsection, we describe the attack approaches used in our experiments. We first give an overview of how to attack a neural network in mathematical notations. Let $x$ be the input to the neural network and $f_{\theta}$ be the function which represents the neural network with parameter $\theta$. The output label of the network to the input can be computed as $c = \arg\max_i f_{\theta}(x)_i$. In order to perform an adversarial attack, we add a small perturbation $\delta_x$ to the original image and get an adversarial image $x_{\text{adv}} = x + \delta_x$. The new input $x_{\text{adv}}$ should look visually similar to the original $x$. Here we use the commonly used $\ell_{\infty}$-norm metric to measure similarity, i.e., we require that $||\delta_x|| \leq \epsilon$. The attack is considered successful if the predicted label of the perturbed image $c_{\text{adv}} = \arg\max_i f_{\theta}(x_{\text{adv}})_i$ is different from $c$.

Generally speaking, there are two types of attack methods: *Targeted Attack*, which aims to change the output label of an image to a specific (and different) one, and *Untargeted Attack*, which only aims to change the output label and does not restrict which specific label the modified example should let the network output.

In this paper, we mainly use the following four gradient-based attack approaches. $J$ denotes the loss function of the neural network and $y$ denotes the ground truth label of $x$.

- **Fast Gradient Sign Method (FGSM).** FGSM (Goodfellow et al., 2015) is a one-step untargeted method which generates the adversarial example $x_{\text{adv}}$ by adding the sign of the gradients multiplied by a step size $\epsilon$ to the original benign image $x$. Note that FGSM

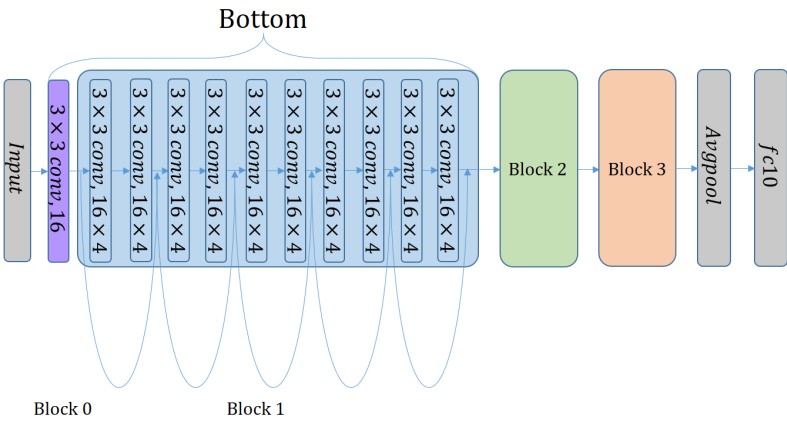

Figure 13: The architecture of WideResNet-32

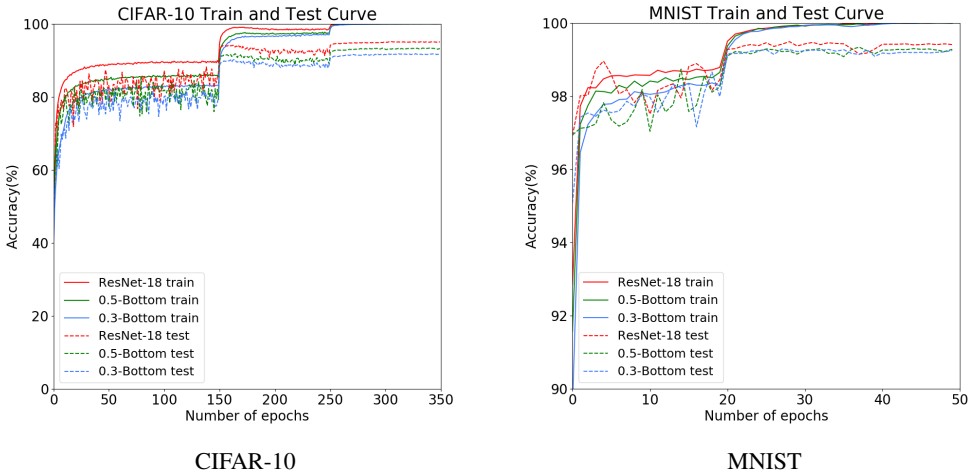

Figure 14: Train and test curve of standard and defective ResNet-18 on CIFAR-10 and MNIST

controls the $\ell_\infty$-norm between the adversarial example and the original one by the parameter $\epsilon$.

$$\boldsymbol{x}_{\text{adv}} = \boldsymbol{x} + \epsilon \cdot \text{sign}(\nabla_{\boldsymbol{x}} J(\boldsymbol{x}, y)).$$

- **Basic iterative method (PGD).** PGD (Kurakin et al., 2016) is a multiple-step attack method which applies FGSM multiple times. To make the adversarial example still stay close to the original image, the image is projected to the $\ell_\infty$-ball centered at the original image after every step. The radius of the $\ell_\infty$-ball is called perturbation scale and is denoted by $\alpha$.

$$\boldsymbol{x}_{\text{adv}}^0 = \boldsymbol{x}, \quad \boldsymbol{x}_{\text{adv}}^{k+1} = \text{Clip}_{\boldsymbol{x}, \alpha} \left[ \boldsymbol{x}_{\text{adv}}^k + \epsilon \cdot \text{sign}(\nabla_{\boldsymbol{x}} J(\boldsymbol{x}_{\text{adv}}^k, y)) \right].$$

- **Momentum Iterative Fast Gradient Sign Method (MIFGSM).** MIFGSM (Dong et al., 2017) is a recently proposed multiple-step attack method. It is similar to PGD, but it computes the optimize direction by a momentum instead of the gradients. The radius of the $\ell_\infty$-ball is also called perturbation scale and is denoted by $\alpha$.

$$g_{k+1} = \mu \cdot g_k + \frac{\nabla_{\boldsymbol{x}} J(\boldsymbol{x}_{\text{adv}}^k, y)}{\|\nabla_{\boldsymbol{x}} J(\boldsymbol{x}_{\text{adv}}^k, y)\|_1}$$

$$\boldsymbol{x}_{\text{adv}}^0 = \boldsymbol{x}, g_0 = 0 \quad \boldsymbol{x}_{\text{adv}}^{k+1} = \text{Clip}_{\boldsymbol{x}, \alpha} \left[ \boldsymbol{x}_{\text{adv}}^k + \epsilon \cdot \text{sign}(g_{k+1}) \right].$$

- **CW Attack.** Carlini & Wagner (2016) shows that constructing an adversarial example can be formulated as solving the following optimization problem:

$$\boldsymbol{x}_{\text{adv}} = \arg\min_{\boldsymbol{x}'} c \cdot g(\boldsymbol{x}') + ||\boldsymbol{x}' - \boldsymbol{x}||_2^2,$$

where $c \cdot g(\boldsymbol{x}')$ is the loss function that evaluates the quality of $\boldsymbol{x}'$ as an adversarial example and the term $||\boldsymbol{x}' - \boldsymbol{x}||_2^2$ controls the scale of the perturbation. More specifically, in the untargeted attack setting, the loss function $g(\boldsymbol{x})$ can be defined as below, where the parameter $\kappa$ is called confidence.

$$g(\boldsymbol{x}) = \max\{\max_{i \neq y}(f(\boldsymbol{x})_i) - f(\boldsymbol{x})_y, -\kappa\},$$

| Architecture | $\text{FGSM}_8$ | $\text{FGSM}_{16}$ | $\text{FGSM}_{32}$ | $\text{PGD}_{16}$ | $\text{PGD}_{2,16}$ | $\text{PGD}_{32}$ | $\text{CW}_{40}$ | Acc |
|---|---|---|---|---|---|---|---|---|
| ResNet-18 | 29.78% | 14.91% | 11.53% | 14.14% | 10.02% | 7.16% | 8.23% | 95.33% |
| 0.7-Bottom | 55.40% | 23.29% | 7.73% | 51.29% | 42.44% | 37.00% | 36.95% | 94.03% |
| 0.5-Bottom | 66.87% | 30.86% | 6.65% | 70.38% | 62.11% | 56.36% | 54.02% | 93.39% |
| 0.3-Bottom | 79.50% | 48.57% | 10.51% | 86.99% | 81.78% | 78.41% | 73.70% | 91.83% |
| 0.25-Bottom | 83.12% | 59.22% | 17.16% | 90.64% | 86.22% | 83.86% | 77.82% | 91.46% |
| 0.2-Bottom | 85.49% | 63.01% | 15.57% | 92.17% | 88.72% | 86.50% | 81.75% | 91.18% |
| 0.15-Bottom | 88.18% | 65.27% | 18.33% | 94.85% | 92.24% | 90.64% | 85.46% | 90.15% |
| 0.1-Bottom | 94.08% | 79.93% | 43.70% | 96.70% | 95.69% | 94.68% | 89.67% | 87.68% |
| 0.05-Bottom | 96.16% | 87.36% | 59.05% | 97.43% | 97.13% | 96.24% | 90.24% | 84.53% |
| 0.7-Top | 28.51% | 14.62% | 8.78% | 10.55% | 7.22% | 4.91% | 7.88% | 95.16% |
| 0.5-Top | 25.01% | 10.76% | 10.24% | 11.06% | 7.99% | 5.10% | 7.19% | 94.94% |
| 0.3-Top | 23.94% | 11.23% | 10.48% | 11.77% | 8.83% | 5.80% | 10.10% | 94.61% |
| 0.5-Bottom, 0.5-Top | 55.88% | 20.77% | 9.96% | 60.75% | 51.47% | 45.29% | 47.32% | 92.48% |
| 0.7-Bottom, 0.7-Top | 51.03% | 24.15% | 10.82% | 45.12% | 35.70% | 30.24% | 29.65% | 94.16% |
| 0.7-Bottom, 0.3-Top | 36.16% | 11.26% | 9.16% | 33.67% | 24.62% | 20.28% | 23.31% | 93.44% |
| 0.3-Bottom, 0.3-Top | 64.85% | 27.43% | 10.09% | 75.49% | 67.09% | 62.78% | 62.47% | 89.78% |
| 0.3-Bottom, 0.7-Top | 74.73% | 40.58% | 9.12% | 82.77% | 75.98% | 72.15% | 68.58% | 91.23% |
| $0.5\text{-Bottom}_{\text{DC}}$ | 36.39% | 12.15% | 8.24% | 19.93% | 14.99% | 11.20% | 12.72% | 95.12% |
| $0.3\text{-Bottom}_{\text{DC}}$ | 43.81% | 17.74% | 8.32% | 27.47% | 21.73% | 16.59% | 19.34% | 94.23% |
| $0.1\text{-Bottom}_{\text{DC}}$ | 49.53% | 19.00% | 7.23% | 53.87% | 44.78% | 41.40% | 44.80% | 93.27% |
| $0.5\text{-Bottom}_{\text{SM}}$ | 77.30% | 48.86% | 12.50% | 85.00% | 80.01% | 75.60% | 72.07% | 92.57% |
| $0.3\text{-Bottom}_{\text{SM}}$ | 82.59% | 48.03% | 12.30% | 91.01% | 87.35% | 84.71% | 79.55% | 89.81% |
| $0.1\text{-Bottom}_{\text{SM}}$ | 67.06% | 39.40% | 16.25% | 80.36% | 74.97% | 72.43% | 65.38% | 74.28% |
| $0.5\text{-Bottom}_{\times 2}$ | 51.25% | 20.78% | 10.29% | 50.16% | 40.47% | 34.24% | 34.00% | 94.12% |
| $0.3\text{-Bottom}_{\times 2}$ | 68.82% | 30.94% | 7.22% | 76.62% | 67.90% | 62.87% | 60.17% | 93.01% |
| $0.1\text{-Bottom}_{\times 2}$ | 88.00% | 68.83% | 28.55% | 93.35% | 90.82% | 88.25% | 82.74% | 90.49% |
| $\text{ResNet-18}_{\text{EN}}$ | 34.98% | 16.51% | 10.32% | 12.60% | 9.22% | 5.48% | 8.46% | 96.03% |
| $0.5\text{-Bottom}_{\times 2,\text{EN}}$ | 58.49% | 20.75% | 8.48% | 57.47% | 47.05% | 39.21% | 41.36% | 95.10% |
| $0.5\text{-Bottom}_{\text{EN}}$ | 69.35% | 31.38% | 7.73% | 75.40% | 66.37% | 60.59% | 58.07% | 94.56% |
| $0.3\text{-Bottom}_{\text{EN}}$ | 81.98% | 51.81% | 8.57% | 90.00% | 85.25% | 82.15% | 77.74% | 93.31% |
| $0.1\text{-Bottom}_{\text{EN}}$ | 95.37% | 81.95% | 43.42% | 97.91% | 97.10% | 95.90% | 91.36% | 89.45% |

Table 11: Extended experimental results of Section 4.3. Adversarial examples generated against *DenseNet-121*. Numbers in the middle mean the success defense rates. The model trained on CIFAR-10 achieves 95.62% accuracy on test set. $p$-Bottom, $p$-Top, $p$-Bottom$_{\text{DC}}$, $p$-Bottom$_{\text{SM}}$, $p$-Bottom$_{\times n}$ and $p$-Bottom$_{\text{EN}}$ mean applying defective layers with keep probability $p$ to bottom layers, applying defective layers with keep probability $p$ to top layers, making whole channels defective with keep probability $p$, using the same defective mask in every channel with keep probability $p$, increasing channel number to $n$ times at bottom layers and ensemble five models with different defective masks of the same keep probability $p$ respectively.

| Architecture | FGSM$_8$ | FGSM$_{16}$ | FGSM$_{32}$ | PGD$_{16}$ | PGD$_{2,16}$ | PGD$_{32}$ | CW$_{20}$ | Acc |
|---|---|---|---|---|---|---|---|---|
| ResNet-18 | 26.99% | 13.91% | 3.57% | 5.98% | 3.70% | 3.02% | 2.19% | 95.33% |
| 0.7-Bottom | 48.76% | 21.32% | 9.54% | 34.43% | 25.14% | 24.16% | 38.87% | 94.03% |
| 0.5-Bottom | 59.66% | 30.48% | 11.60% | 53.89% | 45.47% | 41.27% | 60.65% | 93.39% |
| 0.3-Bottom | 74.00% | 47.11% | 15.65% | 78.23% | 73.30% | 65.83% | 79.04% | 91.83% |
| 0.25-Bottom | 78.37% | 56.05% | 21.44% | 83.96% | 80.09% | 73.45% | 81.59% | 91.46% |
| 0.2-Bottom | 81.67% | 59.14% | 19.60% | 88.18% | 85.07% | 78.72% | 82.78% | 91.18% |
| 0.15-Bottom | 86.31% | 63.16% | 22.23% | 92.26% | 89.99% | 85.14% | 86.06% | 90.15% |
| 0.1-Bottom | 92.89% | 77.90% | 45.63% | 96.27% | 95.30% | 92.80% | 90.29% | 87.68% |
| 0.05-Bottom | 95.07% | 85.40% | 59.91% | 97.51% | 96.69% | 95.30% | 90.97% | 84.53% |
| 0.7-Top | 25.96% | 15.46% | 7.18% | 5.36% | 2.83% | 2.89% | 2.66% | 95.16% |
| 0.5-Top | 25.21% | 9.21% | 1.44% | 5.98% | 4.30% | 3.25% | 3.15% | 94.94% |
| 0.3-Top | 24.36% | 9.49% | 2.60% | 8.54% | 5.30% | 5.02% | 6.62% | 94.61% |
| 0.5-Bottom, 0.5-Top | 51.89% | 20.41% | 10.75% | 45.99% | 37.78% | 34.09% | 52.11% | 92.48% |
| 0.7-Bottom, 0.7-Top | 43.32% | 20.55% | 4.14% | 29.28% | 19.64% | 20.14% | 32.92% | 94.16% |
| 0.7-Bottom, 0.3-Top | 34.09% | 11.05% | 1.58% | 23.06% | 15.26% | 14.93% | 24.11% | 93.44% |
| 0.3-Bottom, 0.3-Top | 61.22% | 28.11% | 13.78% | 67.30% | 59.43% | 52.95% | 69.15% | 89.78% |
| 0.3-Bottom, 0.7-Top | 70.43% | 39.15% | 13.94% | 74.85% | 68.23% | 62.52% | 74.57% | 91.23% |
| 0.5-Bottom$_{DC}$ | 32.86% | 13.89% | 3.71% | 9.41% | 5.60% | 5.34% | 6.10% | 95.12% |
| 0.3-Bottom$_{DC}$ | 37.96% | 16.23% | 5.05% | 16.63% | 11.49% | 10.54% | 15.44% | 94.23% |
| 0.1-Bottom$_{DC}$ | 48.54% | 19.10% | 11.37% | 41.14% | 31.82% | 30.56% | 50.62% | 93.27% |
| 0.5-Bottom$_{SM}$ | 73.96% | 47.63% | 16.60% | 75.60% | 68.88% | 62.10% | 73.68% | 92.57% |
| 0.3-Bottom$_{SM}$ | 80.80% | 48.37% | 15.26% | 87.88% | 84.37% | 77.69% | 82.34% | 89.81% |
| 0.1-Bottom$_{SM}$ | 69.15% | 43.55% | 20.26% | 79.96% | 75.52% | 71.95% | 71.62% | 74.28% |
| 0.5-Bottom$_{\times 2}$ | 46.50% | 21.37% | 6.06% | 32.98% | 22.90% | 22.66% | 39.12% | 94.12% |
| 0.3-Bottom$_{\times 2}$ | 63.37% | 29.90% | 12.07% | 61.81% | 53.25% | 48.36% | 67.02% | 93.01% |
| 0.1-Bottom$_{\times 2}$ | 84.28% | 64.47% | 31.90% | 90.76% | 87.81% | 82.61% | 85.08% | 90.49% |
| ResNet-18$_{EN}$ | 29.36% | 13.89% | 3.81% | 4.72% | 2.88% | 2.08% | 2.09% | 96.03% |
| 0.5-Bottom$_{\times 2,EN}$ | 51.63% | 20.74% | 7.58% | 37.99% | 26.65% | 26.61% | 42.59% | 95.10% |
| 0.5-Bottom$_{EN}$ | 63.38% | 30.25% | 11.05% | 56.29% | 46.76% | 42.75% | 63.90% | 94.56% |
| 0.3-Bottom$_{EN}$ | 77.25% | 50.07% | 13.80% | 80.40% | 75.52% | 68.16% | 80.86% | 93.31% |
| 0.1-Bottom$_{EN}$ | 94.31% | 79.47% | 44.67% | 97.20% | 95.90% | 94.03% | 90.52% | 89.45% |

Table 12: Extended experimental results of Section 4.3. Numbers in the middle mean the success defense rates. Adversarial examples are generated against *ResNet-18*. The model trained on CIFAR-10 achieves 95.27% accuracy on test set. $p$-Bottom, $p$-Top, $p$-Bottom$_{DC}$, $p$-Bottom$_{SM}$, $p$-Bottom$_{\times n}$ and $p$-Bottom$_{EN}$ mean applying defective layers with keep probability $p$ to bottom layers, applying defective layers with keep probability $p$ to top layers, making whole channels defective with keep probability $p$, using the same defective mask in every channel with keep probability $p$, increasing channel number to $n$ times at bottom layers and ensemble five models with different defective masks of the same keep probability $p$ respectively.

| Architecture | $FGSM_8$ | $FGSM_{16}$ | $FGSM_{32}$ | $PGD_{16}$ | $PGD_{2,16}$ | $PGD_{32}$ | $CW_{40}$ | Acc |
|---|---|---|---|---|---|---|---|---|
| ResNet-18 | 29.33% | 15.14% | 3.88% | 0.94% | 1.36% | 0.08% | 0.00% | 95.33% |
| 0.7-Bottom | 45.32% | 18.89% | 9.16% | 12.78% | 13.14% | 3.11% | 1.98% | 94.03% |
| 0.5-Bottom | 56.26% | 27.32% | 10.72% | 33.05% | 31.71% | 15.13% | 8.92% | 93.39% |
| 0.3-Bottom | 70.57% | 42.40% | 14.98% | 67.64% | 65.36% | 48.07% | 33.08% | 91.83% |
| 0.25-Bottom | 77.18% | 53.01% | 19.68% | 77.14% | 74.46% | 59.23% | 39.10% | 91.46% |
| 0.2-Bottom | 80.33% | 56.21% | 18.03% | 83.36% | 81.58% | 69.09% | 47.52% | 91.18% |
| 0.15-Bottom | 84.81% | 61.02% | 21.50% | 89.61% | 87.65% | 78.29% | 53.71% | 90.15% |
| 0.1-Bottom | 92.17% | 77.68% | 45.93% | 94.82% | 94.24% | 90.22% | 66.70% | 87.68% |
| 0.05-Bottom | 94.43% | 85.54% | 60.71% | 96.41% | 96.27% | 93.65% | 71.82% | 84.53% |
| 0.7-Top | 27.78% | 15.03% | 8.07% | 0.60% | 0.82% | 0.00% | 0.00% | 95.16% |
| 0.5-Top | 27.24% | 10.29% | 2.47% | 0.62% | 0.92% | 0.04% | 0.00% | 94.94% |
| 0.3-Top | 24.81% | 9.99% | 2.50% | 0.73% | 1.11% | 0.02% | 0.00% | 94.61% |
| 0.5-Bottom, 0.5-Top | 47.22% | 17.48% | 10.16% | 23.66% | 23.02% | 9.32% | 6.51% | 92.48% |
| 0.7-Bottom, 0.7-Top | 42.18% | 18.20% | 5.26% | 9.88% | 10.52% | 2.41% | 1.64% | 94.16% |
| 0.7-Bottom, 0.3-Top | 33.11% | 11.08% | 2.27% | 6.09% | 6.41% | 0.86% | 0.55% | 93.44% |
| 0.3-Bottom, 0.3-Top | 56.39% | 24.14% | 12.18% | 51.43% | 48.19% | 31.23% | 22.25% | 89.78% |
| 0.3-Bottom, 0.7-Top | 66.33% | 36.31% | 13.09% | 62.31% | 59.74% | 41.88% | 30.68% | 91.23% |
| $0.5\text{-Bottom}_{DC}$ | 31.56% | 13.64% | 4.87% | 1.61% | 1.81% | 0.12% | 0.11% | 95.12% |
| $0.3\text{-Bottom}_{DC}$ | 37.52% | 15.72% | 5.38% | 3.92% | 4.44% | 0.56% | 0.44% | 94.23% |
| $0.1\text{-Bottom}_{DC}$ | 44.00% | 16.90% | 10.30% | 20.34% | 20.32% | 7.95% | 4.95% | 93.27% |
| $0.5\text{-Bottom}_{SM}$ | 69.40% | 41.82% | 14.27% | 62.62% | 60.04% | 40.30% | 26.65% | 92.57% |
| $0.3\text{-Bottom}_{SM}$ | 77.25% | 44.94% | 13.80% | 81.91% | 79.90% | 67.76% | 46.44% | 89.81% |
| $0.1\text{-Bottom}_{SM}$ | 64.32% | 39.76% | 19.21% | 74.47% | 71.88% | 63.05% | 45.18% | 74.28% |
| $0.5\text{-Bottom}_{\times 2}$ | 41.51% | 18.47% | 6.02% | 10.80% | 11.40% | 2.21% | 1.32% | 94.12% |
| $0.3\text{-Bottom}_{\times 2}$ | 58.59% | 25.92% | 11.20% | 42.49% | 40.44% | 21.05% | 13.77% | 93.01% |
| $0.1\text{-Bottom}_{\times 2}$ | 83.05% | 63.73% | 29.22% | 86.47% | 84.39% | 75.07% | 50.11% | 90.49% |
| $ResNet\text{-}18_{EN}$ | 32.80% | 15.67% | 4.65% | 0.70% | 1.00% | 0.02% | 0.00% | 96.03% |
| $0.5\text{-Bottom}_{\times 2, EN}$ | 47.40% | 17.32% | 7.23% | 12.64% | 12.84% | 2.54% | 2.19% | 95.10% |
| $0.5\text{-Bottom}_{EN}$ | 59.64% | 26.21% | 10.17% | 33.55% | 32.11% | 13.93% | 8.12% | 94.56% |
| $0.3\text{-Bottom}_{EN}$ | 73.45% | 45.60% | 12.99% | 69.95% | 67.14% | 48.83% | 32.60% | 93.31% |
| $0.1\text{-Bottom}_{EN}$ | 93.87% | 79.15% | 46.44% | 96.12% | 95.82% | 91.98% | 67.71% | 89.45% |

Table 13: Extended experimental results of Section 4.3. Adversarial examples are generated against *ResNet-50*. Numbers in the middle mean the success defense rates. The model trained on CIFAR-10 achieves 95.69% accuracy on test set. $p$-Bottom, $p$-Top, $p$-Bottom$_{DC}$, $p$-Bottom$_{SM}$, $p$-Bottom$_{\times n}$ and $p$-Bottom$_{EN}$ mean applying defective layers with keep probability $p$ to bottom layers, applying defective layers with keep probability $p$ to top layers, making whole channels defective with keep probability $p$, using the same defective mask in every channel with keep probability $p$, increasing channel number to $n$ times at bottom layers and ensemble five models with different defective masks of the same keep probability $p$ respectively.

| Architecture | FGSM$_8$ | FGSM$_{16}$ | FGSM$_{32}$ | PGD$_{16}$ | PGD$_{2,16}$ | PGD$_{32}$ | CW$_{40}$ | Acc |
|---|---|---|---|---|---|---|---|---|
| ResNet-18 | 25.53% | 17.47% | 8.56% | 3.32% | 3.26% | 1.18% | 0.00% | 95.33% |
| 0.7-Bottom | 46.12% | 23.30% | 10.48% | 33.90% | 29.04% | 19.33% | 2.66% | 94.03% |
| 0.5-Bottom | 57.05% | 31.01% | 11.07% | 57.52% | 51.70% | 40.37% | 14.61% | 93.39% |
| 0.3-Bottom | 72.67% | 48.17% | 15.20% | 82.46% | 77.49% | 71.30% | 39.89% | 91.83% |
| 0.25-Bottom | 78.23% | 58.19% | 21.20% | 87.86% | 84.06% | 77.91% | 47.33% | 91.46% |
| 0.2-Bottom | 82.27% | 61.61% | 19.70% | 91.00% | 87.92% | 83.72% | 51.14% | 91.18% |
| 0.15-Bottom | 85.80% | 65.92% | 22.73% | 94.00% | 91.93% | 88.82% | 57.36% | 90.15% |
| 0.1-Bottom | 92.93% | 79.13% | 48.34% | 96.49% | 95.94% | 94.39% | 65.63% | 87.68% |
| 0.05-Bottom | 94.77% | 87.13% | 63.36% | 97.74% | 97.26% | 95.82% | 69.14% | 84.53% |
| 0.7-Top | 23.76% | 16.66% | 9.52% | 2.39% | 2.37% | 0.84% | 0.00% | 95.16% |
| 0.5-Top | 23.01% | 12.19% | 6.56% | 3.35% | 3.27% | 1.18% | 0.00% | 94.94% |
| 0.3-Top | 22.87% | 11.61% | 6.63% | 5.22% | 4.84% | 1.95% | 0.19% | 94.61% |
| 0.5-Bottom, 0.5-Top | 47.85% | 18.29% | 12.01% | 48.59% | 42.03% | 31.72% | 15.84% | 92.48% |
| 0.7-Bottom, 0.7-Top | 42.34% | 22.07% | 7.84% | 27.19% | 23.31% | 14.06% | 1.70% | 94.16% |
| 0.7-Bottom, 0.3-Top | 31.43% | 12.17% | 6.34% | 19.14% | 15.40% | 8.60% | 1.53% | 93.44% |
| 0.3-Bottom, 0.3-Top | 57.26% | 29.36% | 13.99% | 71.03% | 63.90% | 55.09% | 30.00% | 89.78% |
| 0.3-Bottom, 0.7-Top | 68.66% | 41.32% | 13.72% | 78.61% | 74.22% | 66.02% | 35.71% | 91.23% |
| 0.5-Bottom$_{DC}$ | 30.81% | 14.77% | 6.08% | 6.18% | 5.38% | 2.33% | 0.00% | 95.12% |
| 0.3-Bottom$_{DC}$ | 34.57% | 17.32% | 8.04% | 10.68% | 9.58% | 4.86% | 0.19% | 94.23% |
| 0.1-Bottom$_{DC}$ | 43.46% | 17.61% | 10.54% | 39.53% | 34.10% | 25.01% | 7.41% | 93.27% |
| 0.5-Bottom$_{SM}$ | 71.27% | 49.21% | 16.27% | 80.23% | 74.38% | 65.92% | 34.92% | 92.57% |
| 0.3-Bottom$_{SM}$ | 79.48% | 49.66% | 15.65% | 90.74% | 87.33% | 82.03% | 48.28% | 89.81% |
| 0.1-Bottom$_{SM}$ | 65.85% | 42.59% | 21.87% | 80.36% | 75.71% | 71.25% | 43.22% | 74.28% |
| 0.5-Bottom$_{\times 2}$ | 44.13% | 21.71% | 9.49% | 32.98% | 27.34% | 17.64% | 2.65% | 94.12% |
| 0.3-Bottom$_{\times 2}$ | 60.51% | 30.89% | 11.58% | 66.09% | 59.08% | 48.06% | 21.52% | 93.01% |
| 0.1-Bottom$_{\times 2}$ | 85.26% | 67.91% | 32.51% | 92.46% | 89.99% | 85.86% | 52.96% | 90.49% |
| ResNet-18$_{EN}$ | 27.36% | 17.72% | 8.49% | 2.50% | 2.58% | 0.72% | 0.00% | 96.03% |
| 0.5-Bottom$_{\times 2,EN}$ | 48.07% | 20.83% | 10.35% | 37.11% | 31.01% | 19.68% | 4.91% | 95.10% |
| 0.5-Bottom$_{EN}$ | 60.42% | 31.08% | 10.77% | 60.63% | 54.00% | 41.55% | 13.61% | 94.56% |
| 0.3-Bottom$_{EN}$ | 76.08% | 51.49% | 13.19% | 85.51% | 80.85% | 73.29% | 39.51% | 93.31% |
| 0.1-Bottom$_{EN}$ | 94.40% | 81.32% | 48.52% | 97.58% | 96.87% | 95.33% | 66.99% | 89.45% |

Table 14: Extended experimental results of Section 4.3. Numbers in the middle mean the success defense rates. Adversarial examples are generated against *SENet-18*. The model trained on CIFAR-10 achieves 95.15% accuracy on test set. $p$-Bottom, $p$-Top, $p$-Bottom$_{DC}$, $p$-Bottom$_{SM}$, $p$-Bottom$_{\times n}$ and $p$-Bottom$_{EN}$ mean applying defective layers with keep probability $p$ to bottom layers, applying defective layers with keep probability $p$ to top layers, making whole channels defective with keep probability $p$, using the same defective mask in every channel with keep probability $p$, increasing channel number to $n$ times at bottom layers and ensemble five models with different defective masks of the same keep probability $p$ respectively.

| Architecture | FGSM$_8$ | FGSM$_{16}$ | FGSM$_{32}$ | PGD$_{16}$ | PGD$_{2,16}$ | PGD$_{32}$ | Acc |
|---|---|---|---|---|---|---|---|
| ResNet-18 | 37.67% | 20.25% | 5.40% | 26.97% | 20.65% | 17.58% | 95.33% |
| 0.7-Bottom | 50.06% | 23.54% | 9.53% | 45.27% | 36.74% | 31.61% | 94.03% |
| 0.5-Bottom | 57.35% | 30.52% | 11.13% | 58.66% | 50.82% | 43.89% | 93.39% |
| 0.3-Bottom | 71.75% | 47.35% | 15.47% | 77.57% | 72.68% | 64.06% | 91.83% |
| 0.25-Bottom | 76.81% | 56.69% | 19.44% | 83.32% | 79.23% | 70.72% | 91.46% |
| 0.2-Bottom | 79.46% | 61.45% | 21.36% | 87.55% | 84.41% | 76.35% | 91.18% |
| 0.15-Bottom | 85.51% | 66.55% | 25.35% | 91.65% | 89.29% | 82.90% | 90.15% |
| 0.1-Bottom | 92.58% | 80.68% | 51.90% | 93.41% | 92.83% | 90.30% | 87.68% |
| 0.05-Bottom | 95.24% | 87.10% | 64.22% | 93.14% | 92.78% | 91.14% | 84.53% |
| 0.7-Top | 36.72% | 18.97% | 9.65% | 26.91% | 20.87% | 17.37% | 95.16% |
| 0.5-Top | 35.93% | 13.80% | 2.99% | 26.36% | 21.31% | 17.35% | 94.94% |
| 0.3-Top | 34.05% | 13.06% | 4.04% | 29.84% | 23.08% | 19.04% | 94.61% |
| 0.5-Bottom, 0.5-Top | 50.78% | 19.25% | 9.12% | 52.42% | 45.66% | 38.49% | 92.48% |
| 0.7-Bottom, 0.7-Top | 47.36% | 22.74% | 5.04% | 41.42% | 34.26% | 28.90% | 94.16% |
| 0.7-Bottom, 0.3-Top | 40.38% | 13.50% | 3.28% | 36.96% | 29.80% | 24.98% | 93.44% |
| 0.3-Bottom, 0.3-Top | 59.19% | 28.00% | 12.13% | 68.60% | 62.34% | 53.50% | 89.78% |
| 0.3-Bottom, 0.7-Top | 67.14% | 40.57% | 13.80% | 73.93% | 69.07% | 60.58% | 91.23% |
| 0.5-Bottom$_{DC}$ | 37.37% | 16.99% | 6.62% | 26.39% | 20.09% | 16.79% | 95.12% |
| 0.3-Bottom$_{DC}$ | 42.39% | 19.90% | 6.74% | 28.85% | 22.83% | 20.22% | 94.23% |
| 0.1-Bottom$_{DC}$ | 47.41% | 21.12% | 11.43% | 45.20% | 38.32% | 32.68% | 93.27% |
| 0.5-Bottom$_{SM}$ | 69.61% | 46.57% | 14.85% | 76.26% | 70.62% | 61.99% | 92.57% |
| 0.3-Bottom$_{SM}$ | 79.69% | 48.86% | 13.87% | 87.03% | 83.43% | 76.01% | 89.81% |
| 0.1-Bottom$_{SM}$ | 67.77% | 44.38% | 20.74% | 68.67% | 66.66% | 61.88% | 74.28% |
| 0.5-Bottom$_{\times 2}$ | 46.93% | 21.74% | 7.11% | 45.08% | 37.12% | 31.21% | 94.12% |
| 0.3-Bottom$_{\times 2}$ | 60.23% | 29.72% | 11.07% | 63.62% | 57.28% | 48.83% | 93.01% |
| 0.1-Bottom$_{\times 2}$ | 83.32% | 66.44% | 33.11% | 89.57% | 87.33% | 80.20% | 90.49% |
| ResNet-18$_{EN}$ | 39.75% | 18.73% | 6.59% | 26.87% | 20.33% | 17.22% | 96.03% |
| 0.5-Bottom$_{\times 2,EN}$ | 51.91% | 19.60% | 7.86% | 49.29% | 39.53% | 34.29% | 95.10% |
| 0.5-Bottom$_{EN}$ | 60.43% | 31.07% | 10.50% | 61.60% | 53.91% | 46.50% | 94.56% |
| 0.3-Bottom$_{EN}$ | 74.11% | 50.89% | 13.54% | 80.75% | 76.27% | 66.71% | 93.31% |
| 0.1-Bottom$_{EN}$ | 94.14% | 82.46% | 52.59% | 96.22% | 95.20% | 92.64% | 89.45% |

Table 15: Extended experimental results of Section 4.3. Numbers in the middle mean the success defense rates. Adversarial examples are generated against *VGG-19*. The model trained on CIFAR-10 achieves 94.04% accuracy on test set. $p$-Bottom, $p$-Top, $p$-Bottom$_{DC}$, $p$-Bottom$_{SM}$, $p$-Bottom$_{\times n}$ and $p$-Bottom$_{EN}$ mean applying defective layers with keep probability $p$ to bottom layers, applying defective layers with keep probability $p$ to top layers, making whole channels defective with keep probability $p$, using the same defective mask in every channel with keep probability $p$, increasing channel number to $n$ times at bottom layers and ensemble five models with different defective masks of the same keep probability $p$ respectively.

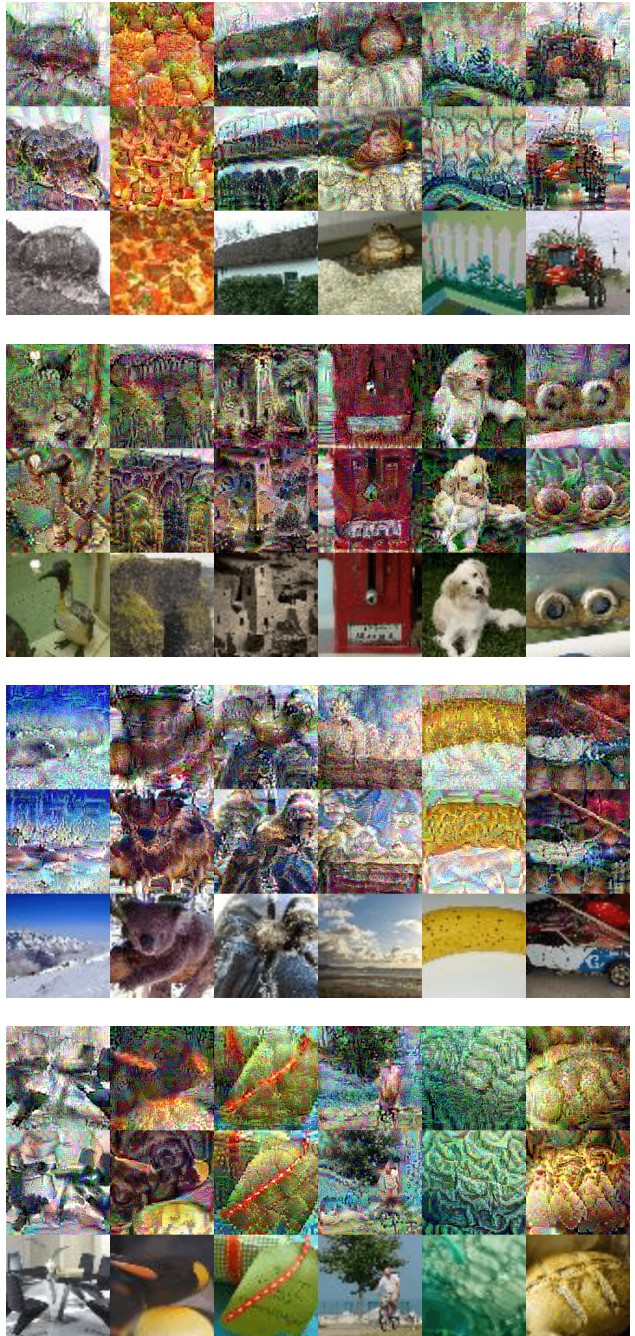

Figure 15: Randomly sampled images from Tiny-ImageNet dataset. The network structure used to generate these images is defective ResNet-18 with keep probability $0.1$ on the $1^{st}, 2^{nd}$ blocks. The attack method is MIFGSM with perturbation scale $64$, step size $1$ and step number $80$. For each image, we show the image generated against network with Random Mask (upper), the image generated against the ResNet-18 (middle) and the original image (lower).

