# OpenReview forum: "Defective Convolutional Layers Learn Robust CNNs"
_ICLR.cc/2020/Conference — Reject_

### Official Review · AnonReviewer3 · 2019-10-17
**Official Blind Review #3**

**Rating:** 3

**Review:**

The paper deals with robustness against adversarial attacks. It proposes to blank out large parts of the early convolution layers in a CNN, in an attempt to shift the focus from "texture" to "shape" features. This does seem to improve robustness against adversarial examples, with only a small decrease in general classification performance. The explanation for this, on the other hand, is not really convincing.

The idea is simple: adversarial noise introduces high-frequency texture patterns, so destroy those by blanking out a large portion of the neurons in a layer. Quite obviously, this can have an influence - when blanking out 90% of the pixels (as suggested in the paper), the effective sampling resolution goes down by a factor of 3 in each axis, and high-frequency patterns are a lot less likely to be picked up. It does, however, remain unclear why this approach is particularly useful, or why it even works at all. On the one hand, an easier way to surely get rid of those patterns is simply to blur the images accordingly before feeding them to the network. That baseline is missing. On the other hand it is quite an outrageous claim that one can throw away 90% of the responses in the early conv layers with hardly any performance loss. I don't doubt the experiments results, but if one discovers something like that, it needs to be explained. The network has  a lot lessless capacity, effectively loses a factor 3 in resolution, but performance seemingly stays almost the same!

This brings me to another point. It is never really defined what is meant by "texture" respectively "shape". By reverse-engineering from the paper text I gather that "shape" is simply texture at a significantly lower resolution. But then how does destroying "texture" affect objects with significantly lower size/scale in the image?

A few technical questions remain unclear. First, the adversarial noise in the paper looks a lot stronger than normal. It is easily visible, and in that sense not "adversarial". In fact the paper openly states that they "even fool humans". So since the labels are human-annotated, these are in fact not adversarial examples of class A, but examples of a different class wrongly labeled as class A...

Another question is how much "texture" is lost, since the paper finds it important to use a different random mask for each filter in a layer. So does that really suppress so much texture? Almost all pixels will be seen by some non-defective filters, so it would seem that the hi-res information is implicitly still there. To really suppress texture it would seem more effective to always use the same mask, but that apparently does not work. Why?

What completely confused me was which networks were actually used to generate the adversarial examples. Are these adversarial against the standard CNN or against the defective one? If defective convolutions indeed become popular, then an attacker would obviously know about that and also use a defective network to generate his adversarial examples.

One thing I did not understand is the incoherent mixture of datasets. The first experiment with the reshuffled tiles is done on ImageNet. But then the actual experiments regarding robustness against attacks are done only with Cifar-10. Why suddenly switch? And then, many of the visual examples are from TinyImageNet. Why switch again? And on that account, since apparently all of them were processed, is the behaviour consistent across datasets?

A note aside, I am not sure it is a good idea to treat additive Gaussian noise in the same way as adversarial patterns. Some level of noise that is at least approximately Gaussian is present in almost all images. So it is actually a good thing if a network learns the magnitude of that noise, so as to separate it from the signal, i.e., the brightness variations that are informative and not Gaussian noise. In that view it is a good thing, not a weakness, if adding noise of the wrong magnitude misleads the network (although, ideally, it should of course flag the image as being out-of-distribution).

In summary, I find the results interesting - in particular also the one on tiled and reshuffled images. But I am at this point not convinced by the explanations. If one can indeed throw away 90% of all responses in the low layers then that would be a rather big thing that needs an explanation. Unless the task is easy enough to be solved with 3x lower resolution - but in that case I would expect that simply reducing the resolution would also destroy the adversarial pattern. I am on the fence, but  in its current state the paper leaves too many open questions.


**Experience Assessment:**

I have read many papers in this area.

**Review Assessment: Checking Correctness Of Derivations And Theory:**

I assessed the sensibility of the derivations and theory.

**Review Assessment: Checking Correctness Of Experiments:**

I assessed the sensibility of the experiments.

**Review Assessment: Thoroughness In Paper Reading:**

I read the paper at least twice and used my best judgement in assessing the paper.

---

> ### Author Response · Authors · 2019-11-15
> **Response to Reviewer #3**
>
> We thank Reviewer #3 for the thorough analysis and suggestions. We will try our best to address the concerns and are open to further discussions.
>
>
> Regarding the one can throw away 90% of the responses in the early conv layers with hardly any performance loss and simply reducing the resolution may also destroy the adversarial pattern:
>
> We also find this point very surprising that freezing quite a lot of neurons in the network would not drastically degrade the performance, and validated it in five architectures (DenseNet-121, SENet18, VGG-19, ResNet-18, ResNet-50). To ensure this, we attach the codes in revision. Please refer to the link and have a check.
>
> Besides, there are three points of our findings that may help us understand this point.
> The number of parameters in the network is not reduced after including defective neurons.
> To obtain high test accuracy, it is very important to introduce various patterns for defective neurons arrangement across channels. According to Table 4, if the patterns are the same across channels, the test accuracy will significantly drop (from 0.1-Bottom 87.7 to 0.1-Bottom_{SM} 74.3).
> According to Table 4, applying the method on Bottom layers are sensitive than on Top layers about the test accuracy.
>
> During rebuttal, we downsample the images in CIFAR-10 to 8x8 and find the test accuracy will drop to 77.3% which is a very low value. Some related works ([1], [2]) have studied the effects of input transformations on adversarial robustness.  According to their results, those transformations will indeed destroy the adversarial pattern to some degree but also decrease the test accuracy a lot.
>
> [1] Countering Adversarial Images using Input Transformations, https://arxiv.org/abs/1711.00117
> [2] Feature Squeezing: Detecting Adversarial Examples in Deep Neural Networks, https://arxiv.org/abs/1704.01155
>
>
>
>
> Regarding if defective convolutions indeed become popular, then an attacker would obviously know about that and also use a defective network to generate his adversarial examples:
>
> We have considered this point and conducted thorough experiments about the gray-box setting in Appendix A.7. For the proposed method, we actually found there are two kinds of gray-box settings. One way is to generate adversarial examples against one trained neural network and test those images on a network with the same structure but different initializations. The other way is specific to our defective models. We generate adversarial examples on one trained defective CNN and test them on a network with the same keep probability but different samplings of defective neurons. The results show the proposed model outperforms than standard models in the gray-box setting and, thus, alleviate the mentioned issues.
>
>
>
>
> Regarding the additive Gaussian noise:
>
> According to Figure 5, networks actually perform very well when the scale of Gaussian noise is small, which is aligned with your analysis. Indeed, it is not a weakness for networks cannot resist the Gaussian noise if adding noise of the wrong magnitude. But, as the noise scale changing from 0/255 to 64/255 in \ell_\infty, we can clearly see the performance of the proposed model is consistently higher than the standard one.  We believe this phenomenon is non-trivial. In addition, [3] uses empirical evidence to points out that a successful adversarial defend method should also effectively defense against images with additive Gaussian noise.
>
> [3] Adversarial examples are a natural consequence of test error in noise, https://arxiv.org/abs/1901.10513
>
>
>
>
> Regarding the adversarial noise in the paper looks a lot stronger than normal:
>
> The noise for the visual examples of CIFAR-10 is 8/255 and 16/255 in \ell_\infty. The noise for the visual examples of Tiny-ImageNet is 16/255 and 32/255 in \ell_\infty. You can check this point by directly crop the images from the paper and compare it with the original images. We believe those scales are commonly used in the related papers. The reason why noise looks stronger than normal is probably that our model finds a suitable direction for those noises.
>
>
>
>
> Regarding these are in fact not adversarial examples of class A, but examples of a different class wrongly labeled as class A:
>
> Please refer to Figure 1, the images in the first row are generated from the images in the second row. The names below the first row are the predicted class by our models, and the names below the second row are the ground-truth class. All the ground-truth labels are correct. The new predicted labels are corresponding to the images that demonstrate our claim that some adversarial examples generated by our model can even fool humans.

---

> > ### Author Response · Authors · 2019-11-15
> > **Response to Reviewer#3 [cont'd]**
> >
> > Regarding the “texture” and “shape”, and how does destroying "texture" affect objects with significantly lower size/scale in the image.
> >
> > Thanks for pointing this out. In our paper, the shape means the boundaries of objects and the texture means the repeated patterns over objects. It is hard to explicitly find the effects of the small objects with destroying "texture". But, from the results of random shuffling experiments, you might also think such destroying would affect the recognization of small objects. Take Figure 3 as an example, our humans can still recognize the Ailurus fulgens according to the noses and ears, but the model would fail in this case.
> >
> >
> >
> >
> > Regarding the suppress texture it would seem more effective to always use the same mask, but that apparently does not work:
> >
> > According to Table 4, the 0.5-Bottom_{SM} indeed outperforms 0.5-Bottom, which is aligned with the analysis. But the 0.1-Bottom_{SM} is worse than 0.1-Bottom. We indicate that the randomness in generating masks in different channels involves various topological structures for local feature extraction instead of learning. When the keep probability is very low, the way of using the same mask across channels will result in limited topological structures for local feature extraction.
> >
> >
> >
> >
> > Regarding which networks were actually used to generate the adversarial examples:
> >
> > We generate adversarial examples against the standard CNNs. For each experiment, we illustrated the specific setting in the corresponding descriptions.
> >
> >
> >
> >
> > Regarding the incoherent mixture of datasets:
> >
> > In our paper, we mainly show the results on CIFAR-10 and show the qualitative results on both CIFAR-10 and Tiny-ImageNet. To better evaluate our methods, we also perform experiments on MNIST and ImageNet. The inconsistent mainly due to the compared methods and computation cost. The improvements of robustness are consistent across datasets. But currently, we can not find the visual samples that can fool humans on ImageNet trained models.

---

### Official Review · AnonReviewer2 · 2019-10-22
**Official Blind Review #2**

**Rating:** 1

**Review:**

This paper presents a technique for adversarial defense, employing what the authors refer to as "defective  convolution layers".  It attempts to use such layers to provide more adversarially robust models. The paper is easy to follow and well written, but the experimentation is lacking, and so the contribution is limited.

Defective  Convolution  layers make use of a random fixed masking matrix, to fix some number of neurons to constant values, effectively removing the incoming weights of these neurons from the optimization problem. The outgoing weights of the masked neurons are effectively additional bias terms for the layer above. Applying this technique to a fully connected layer would be equivalent to training with a fully connected layer with a smaller size.

The paper presents the accuracy of such models on a variety of black box attacks. They focus on black box for two reasons, neither of which are particularly convincing. They claim that the white box attacks are semantically meaningful and so could fool a human, but no human evaluation is presented and the examples which are illustrated do not demonstrate this property.  They also claim to focus on black box attacks because is it more practical in a real world setting. However the model they present achieves significantly lower test accuracy on clean data than a standard network, so the practical deployment of such a model seems unlikely in its current implementation.

The paper presents thorough ablation studies on the architectural choices that go into this model. This is a positive quality of the work. However they do not compare standard networks with similar test accuracy to their defective models. As they note in the paper and in the appendix, there is a correlation between test accuracy and adversarial robustness. Based on the findings which they present, it is unclear if the effect of their "Defective Convolution Layer" is simply to reduce the test accuracy and thereby increase the adversarial robustness. This issue must be addressed for the work to contribute to the adversarial literature in a meaningful way.

**Experience Assessment:**

I have published one or two papers in this area.

**Review Assessment: Checking Correctness Of Derivations And Theory:**

N/A

**Review Assessment: Checking Correctness Of Experiments:**

I assessed the sensibility of the experiments.

**Review Assessment: Thoroughness In Paper Reading:**

I read the paper at least twice and used my best judgement in assessing the paper.

---

> ### Author Response · Authors · 2019-11-15
> **Response to Reviewer #2**
>
> We thank Reviewer #2 for the feedback and suggestions. The suggestions are helpful, and most concerns are already considered in the paper.
>
>
> Regarding the proposed method is simply to reduce the test accuracy and thereby increase the adversarial robustness:
>
> Please refer to the first paragraph on Page 7. To eliminate this point, all numbers in our paper except Table 2 are calculated on the adversarial examples whose corresponding original images can be classified correctly by the tested model.
>
>
>
>
> Regarding the model they present achieves significantly lower test accuracy on clean data than a standard network:
>
> We want to emphasize that the proposed method has the state-of-the-art performance on transfer-based black-box defense while maintaining the highest test accuracy. In our experiments, we also demonstrate the effectiveness of the proposed method against the decision-based attack which is a more realistic setting.
>
>
>
>
> Regarding the correlations between test accuracy and adversarial robustness:
>
> The phenomena are specific to the proposed method. Different keep probabilities would bring different improvements on the robustness and declinations on test accuracy. It is hard to apply similar experiments on standard models.
>
>
>
>
> Regarding the claim that the white box attacks are semantically meaningful and so could fool a human, but no human evaluation is presented and the examples which are illustrated do not demonstrate this property:
>
> This work is an attempt to discover a kind of CNNs which learn features that are more aligned with human perception. So far, there are no works can achieve better or the same effects on CIFAR-10 and Tiny-ImageNet by modifying the architecture of CNNs. Please compare the generated adversarial examples along with their corresponding predicted classes.

---

### Official Review · AnonReviewer1 · 2019-10-24
**Official Blind Review #1**

**Rating:** 6

**Review:**

* Summary *
The paper proposes defective convolutional layers as a measure of defense against adversarial attacks on deep neural networks. This layer sets the outputs of a randomly sampled but *fixed* set of neurons in the convolutional layers to zero during training and testing. The authors claim that defective convolutional layers encourage the model to pick up features other than local textures, e.g. shape information. The shape-vs-texture tradeoff is supported by experiments showing that defective CNNs perform worse than normal CNNs on images with permuted patches and that adversarial examples with larger epsilons exhibit more semantic shapes. The detailed experiment section evaluates the method on transfer-based, gray-box and black-box adversarial attacks,	including Gaussian noise. Additionally, it provides ablation studies on the keep-probability and position of the defective layer.

Overall I think this is a valuable contribution to a topic of high interest for ICLR and should be accepted. The method is simple to implement, has minor impact on the test accuracy and seems to increase robustness measures under the proposed settings across all of the tested architectures. However, the evaluation is lacking w.r.t. natural robustness, more detailed evaluation on the gray-box, black-box and white-box attacks. In the white-box setting (Table 8) a stronger attacker with a larger number of iterations and random restarts should be used in order to ensure that the difference in defense performance is real.

The experiment section should have a stronger focus on the gray-box attacks where the source network also has defective layers, since the method alters the network architecture and presumably learn a different set of features. However, the lack of transfer from "normal models",	can also be seen as supporting argument for the model picking up different, potentially robust, features, following the argument in [4].

Because the idea is motivated from the texture-vs-shape discussion [1], an evaluation on natural/empirical robustness under image corruptions [2], e.g. CIFAR10-C or ImageNet-C, and/or comparison	to a network trained on Stylized-ImageNet, should be conducted.


Additional Feedback:
- Since the method is closer to the original (neuron-wise) dropout than to SpatialDropout and DropBlock, including this in the evaluation would be appreciated.
- To show the alignment of the adversarial gradients with the human-vision, the authors could visualize the loss gradients similar to [3] (Figure 2b)
- For comparison in A.4, the median squared L2-distance of the adversarially trained Madry model under the decision-based attack would be helpful.
- Why is the sampling of 4000 images in the patch experiment 3.2 done? What happens to the images that are correctly predicted but with <99% confidence?
- What happens if this method is combined with adversarial training?

[1] ImageNet-trained CNNs are biased towards texture; increasing shape bias improves accuracy and robustness, https://arxiv.org/abs/1811.12231
[2] Benchmarking Neural Network Robustness to Common Corruptions and Surface Variations, https://arxiv.org/abs/1807.01697
[3] Robustness May Be at Odds with Accuracy, https://arxiv.org/abs/1805.12152
[4] Adversarial Examples Are Not Bugs, They Are Features, https://arxiv.org/abs/1905.02175

**Experience Assessment:**

I have read many papers in this area.

**Review Assessment: Checking Correctness Of Derivations And Theory:**

I assessed the sensibility of the derivations and theory.

**Review Assessment: Checking Correctness Of Experiments:**

I assessed the sensibility of the experiments.

**Review Assessment: Thoroughness In Paper Reading:**

I read the paper thoroughly.

---

> ### Author Response · Authors · 2019-11-15
> **Response to Reviewer #1**
>
> We thank Reviewer #3 for the feedback and suggestions. The suggestions are really helpful in further improving our work.
>
> Regarding sampling of 4000 images in the patch experiment with confidence > 99%:
>
> The phenomenon is similar between the images that are correctly predicted with confidence score > 99% and < 99%. We set that threshold for reducing the effects of test accuracy.
>
>
>
>
> Regarding the adversarial training:
>
> During the rebuttal, we adversarially trained a defective CNN under the same setting described in [1] and reach 51.6% successful defense rate against the default PGD attack (\ell_\infty = 8 and 7 steps) used in training, which outperforms the standard CNN (50.0%). And the test accuracy will drop from 87.3% to 83.3%.
>
> [1] Towards Deep Learning Models Resistant to Adversarial Attacks, https://arxiv.org/abs/1706.06083
>
>
>
>
> Regarding the experiments on ImageNet-C, Stylized-ImageNet and etc:
>
> Thanks for the suggestion. In our paper, we want to prove this point by random shuffling experiments. We believe adding such experiments will strong the conclusions.
>
>
>
>
> Regarding the experiments of the original (neuron-wise) dropout:
>
> Since there are fewer people would adopt this kind of dropout into CNNs, we choose to presents the results of two popular ways. We will seriously consider this point in revision.
>
>
>
>
> Regarding a stronger attacker with a larger number of iterations and random restarts on white-box, more thorough  gray-box experiments, and visualizing the loss gradients:
>
> Thanks for those valuable suggestions. We really appreciate it and will conduct them to strengthen the paper.

---

### Decision · Program_Chairs · 2019-12-19

**Decision:**

Reject

**Comment:**

The reviewers wondered about the practical application of this method, given that the performance was lower.  The reviewers were also surprised by some of your claims and wanted you to explore them more deeply.

On the positive side, the reviewers found your experiments to be very thorough.  You also performed additional experiments during the rebuttal period.  We hope that those experiments will help you to build a better paper as you work towards publishing this work.